# Evolutionary rescue of spherical *mreB* deletion mutants of the rod-shape bacterium *Pseudomonas fluorescens* SBW25

**Paul Richard J Yulo**[1†], **Nicolas Desprat**[2,3,4*†], **Monica L Gerth**[5‡],
**Barbara Ritzl-Rinkenberger**[6,7], **Andrew D Farr**[8], **Yunhao Liu**[5], **Xue-Xian Zhang**[1],
**Michael Miller**[1], **Felipe Cava**[6,7], **Paul B Rainey**[5,8,9*], **Heather L Hendrickson**[1,10*]

[1]Institute of Natural and Mathematical Science, Massey University, Auckland, New Zealand; [2]Laboratoire de Physique de l'ENS, Université Paris Cité, Ecole normale supérieure, UniversitéPSL, Sorbonne Université, CNRS, 75005 Paris, Paris, France; [3]Institut de biologie de l'Ecole normale supérieure (IBENS), Ecole normale supérieure, CNRS, INSERM, PSL Research University, Paris, France; [4]Université Paris Cité, Paris, France; [5]New Zealand Institute for Advanced Study, Massey University, Auckland, New Zealand; [6]Department of Molecular Biology, Umeå University, Umeå, Sweden; [7]Laboratory for Molecular Infection Medicine Sweden, Umeå Centre for Microbial Research, SciLifeLab, Umeå Centre for Microbial Research, Umeå University, Umeå, Sweden; [8]Department of Microbial Population Biology, Max Planck Institute for Evolutionary Biology, Plön, Germany; [9]Laboratoire Biophysique et Évolution, CBI, ESPCI Paris, Université PSL, Paris, France; [10]School of Biological Sciences, University of Canterbury, Christchurch, New Zealand

*For correspondence:
nicolas.desprat@phys.ens.fr (ND);
rainey@evolbio.mpg.de (PBR);
heather.hendrickson@canterbury.ac.nz (HLH)

†These authors contributed equally to this work

**Present address:** ‡School of Biological Sciences, Victoria University of Wellington, Wellington, New Zealand

## eLife Assessment

This **important** study combines **convincing** evolution experiments with molecular and genetic techniques to study how a genetic lesion in MreB that causes rod-shaped cells to become spherical, with concomitant deleterious fitness effects, can be rescued by natural selection. The detailed mechanistic investigation increases our understanding of how mreB contributes to cell wall synthesis and shows how compensatory mutations may reestablish its homogeneity.

**Abstract** Maintenance of rod-shape in bacterial cells depends on the actin-like protein MreB. Deletion of *mreB* from *Pseudomonas fluorescens* SBW25 results in viable spherical cells of variable volume and reduced fitness. Using a combination of time-resolved microscopy and biochemical assay of peptidoglycan synthesis, we show that reduced fitness is a consequence of perturbed cell size homeostasis that arises primarily from differential growth of daughter cells. A 1000-generation selection experiment resulted in rapid restoration of fitness with derived cells retaining spherical shape. Mutations in the peptidoglycan synthesis protein Pbp1A were identified as the main route for evolutionary rescue with genetic reconstructions demonstrating causality. Compensatory *pbp1A* mutations that targeted transpeptidase activity enhanced homogeneity of cell wall synthesis on lateral surfaces and restored cell size homeostasis. Mechanistic explanations require enhanced understanding of why deletion of *mreB* causes heterogeneity in cell wall synthesis. We conclude by presenting two testable hypotheses, one of which posits that heterogeneity stems from

non-functional cell wall synthesis machinery, while the second posits that the machinery is functional, albeit stalled. Overall, our data provide support for the second hypothesis and draw attention to the importance of balance between transpeptidase and glycosyltransferase functions of peptidoglycan building enzymes for cell shape determination.

## Introduction

The rescue of fitness-compromised mutants by selection is a useful strategy to obtain new understanding into the molecular determinants of complex phenotypes (*Remigi et al., 2019*; *LaBar et al., 2020*). The approach has shed light on factors such as the flexibility of regulatory networks (*Lind et al., 2015*), the diversity of structural components affecting ecological success (*Lind et al., 2017*), the predictability of evolution (*Lind et al., 2019*), the origins of new genes (*Näsvall et al., 2012*), and the causes of changes in interactions among bacteria and their hosts (*Remigi et al., 2014*). Here, we turn attention to the evolutionary rescue of a *Pseudomonas fluorescens* mutant lacking MreB, a key component of cell shape (*Errington, 2003*) that is conserved in a broad range of rod-shaped bacteria (*Shi et al., 2018*).

For model organisms, notably, *Escherichia coli* and *Bacillus subtilis*, the molecular mechanisms underpinning cell shape have been the subject of intensive investigation (*Shi et al., 2018*; *van Teeffelen and Renner, 2018*; *Egan et al., 2020*). The primary determinant of bacterial cell shape in both Gram positive and Gram negative cells is peptidoglycan. Its synthesis and assembly is directed by two main multiprotein complexes that each comprise a combination of enzymes and regulatory elements involved in cell wall dynamics, DNA segregation, and cell division (*Daniel and Errington, 2003*; *Young, 2003*; *Osborn and Rothfield, 2007*; *Young, 2007*; *Huang et al., 2008*; *Young, 2010*). The first of these, the elongasome (or rod complex), inserts newly synthesised peptidoglycan strands into the lateral wall of the growing cell, while the divisome participates in septum positioning and cell wall constriction (*van Teeffelen and Renner, 2018*). Both elongasome and divisome rely on penicillin-binding proteins (PBPs) to polymerise (transglycosylate) and/or to cross-link (transpeptidate) newly synthesised peptidoglycan strands.

The elongasome consists of several catalytically interacting components including the transglycosylase (TGase) RodA (*Cho et al., 2016*), the transpeptidase (TPase) Pbp2 (*Ishino et al., 1986*), that function in concert with class A PBP (aPBP), Pbp1A, which has both TGase and TPase activities (*Randich and Brun, 2015*). Activity of Pbp1A, while not strictly a member of the elongasome, depends on the outer membrane-anchored protein LpoA (*Sardis et al., 2021*), with Pbp1A interacting with Pbp2 (*Banzhaf et al., 2012*; *Egan et al., 2020*; *Kang and Boll, 2022*). Central to elongasome function – and essential for construction of rod-shaped cells – is the actin-like protein MreB. MreB forms helical-like structures tethered to the inner membrane and coordinates the pattern of new wall growth along the lateral part of the cell cylinder through interactions with other components of the elongasome (*Typas et al., 2011*; *Shi et al., 2018*). Indicative of a balance between diffusive synthesis affected by class A PBPs and processive activities of the elongasome is the fact that deletion of class A PBPs leads to up-regulation of rod complex activity (*Patel et al., 2020*), with evidence suggesting that the two systems work semi-autonomously (*Cho et al., 2016*).

Most bacteria contain a second aPBP, Pbp1B, which, like Pbp1A, has both TGase and TPase activity. The two aPBPs are partially redundant (*Yousif et al., 1985*), but based on localisation of Pbp1B to the mid-cell during division (*Bertsche et al., 2006*), the standard view is that Pbp1B contributes to divisome function (particularly through interaction with PBP3), whereas Pbp1A contributes to cell elongation (*Egan et al., 2020*). However, it is apparent that Pbp1B also contributes to elongation (*Typas et al., 2011*; *Cho et al., 2016*), with evidence from *E. coli* that its role may be more important than Pbp1A (*Vigouroux et al., 2020*). In particular it was recently shown that Pbp1B responds to cell wall damage by repairing defects (*Vigouroux et al., 2020*) and that the bound (enzymatically active) form of Pbp1B is elevated in the absence of Pbp1A (*Vigouroux et al., 2020*).

In both *E. coli* and *B. subtilis* (and most other rod-shaped bacteria), MreB is required for maintenance of rod-shape (*Shi et al., 2018*). While deletion of the gene is lethal under normal growth conditions, inhibition of MreB function by depolymerisation via the drug A22 causes abnormal cell wall growth leading to spherical cells that swell and eventually lyse (*Iwai et al., 2002*). An earlier study in *B. subtilis* – where cells devoid of MreB were maintained in a viable state by culture in the presence

of magnesium – found that transposon-induced mutations in *pbp1A* could rescue cell viability in the absence of magnesium, but could not return cells to rod-shape (*Kawai et al., 2009*).

Here, with interest in evolution of the machinery underpinning cell shape, we took advantage of prior knowledge that inactivation of *mreB* in the rod-shaped bacterium *P. fluorescens* SBW25, while reducing fitness, is not lethal (*Spiers et al., 2002*), and asked whether selection could compensate for this cost. Of particular interest was: (1) the capacity for compensatory evolution to restore fitness; (2) the number, timing, and impact of fitness-restoring mutations; and (3) the connection between fitness-restoring mutations, growth dynamics, and cell morphology.

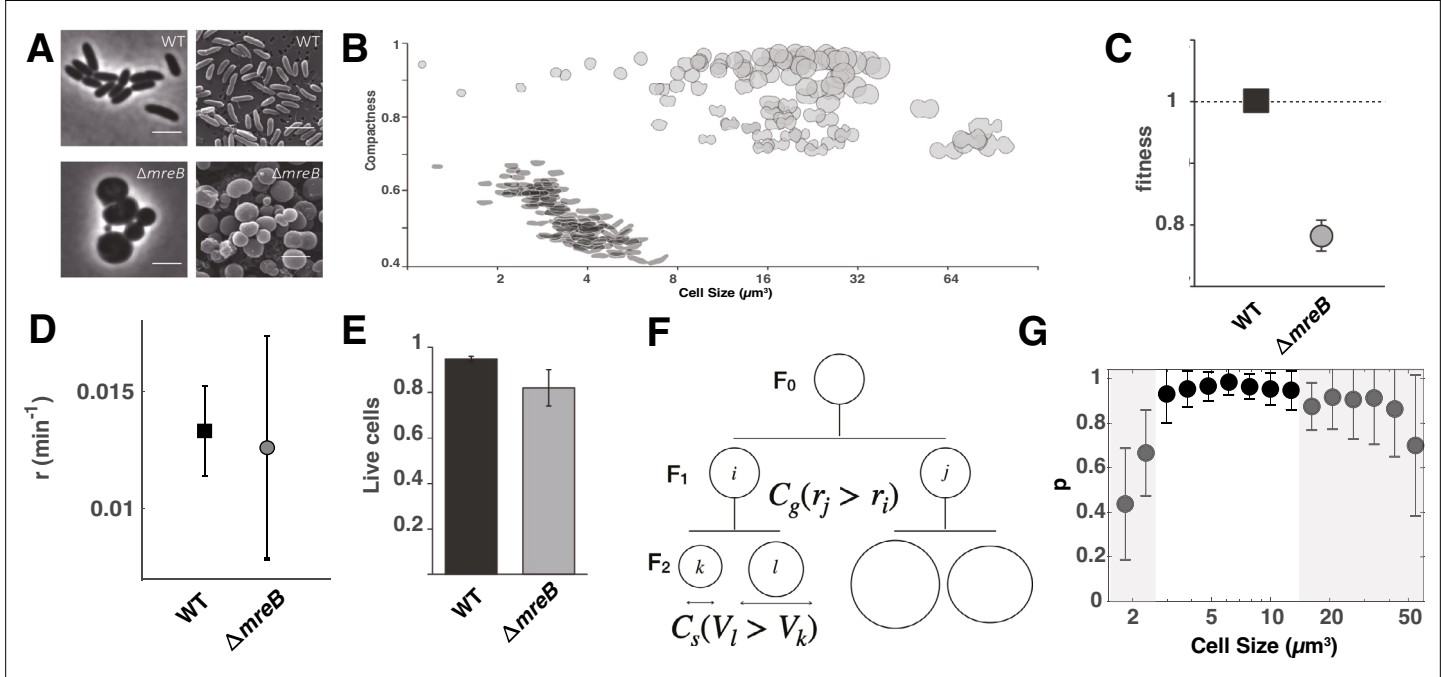

**Figure 1.** Characterisation of ancestral SBW25 and Δ*mreB* strains. Figure shows (**A**) photomicrographs of ancestral SBW25 and Δ*mreB*. Scale bars, 3 µm. (**B**) The relationship between cell shape and estimated volume ($V_e$) is represented using compactness, a measure of roundness. One hundred representative cells from ancestral SBW25 and Δ*mreB* populations are shown as cell outlines. (**C**) Fitness of ancestral SBW25 GFP and the Δ*mreB* mutant relative to ancestral SBW25 when both are in exponential phase during pairwise competition assays. Data are means and standard deviation of 50,000 cells each of ancestral SBW25 and Δ*mreB*, respectively, and 100,000 cells total, after competition. (**D**) Elongation rate measured for SBW25 and Δ*mreB* SBW25. Error bars represent mean and standard deviation ($N_{WT}$ = 94; $N_{\Delta mreB}$ = 99). (**E**) Proportion of live cells in ancestral SBW25 (black bar) and Δ*mreB* (grey bar) based on LIVE/DEAD BacLight Bacterial Viability Kit protocol. Cells were pelleted at 2000 × *g* for 2 min to preserve Δ*mreB* cell integrity. Error bars are means and standard deviation of three biological replicates (n>100). (**F**) Schematic of asymmetric size production in cell lineages at generation F0 and F1. Asymmetries from differential growth rate between sister cells (*i,j*) are captured by $C_g$ (see Materials and methods), while asymmetries in daughter (*l,k*) cell sizes at septation are measured by $C_s$. (**G**) Probability p to pass to the next generation as a function of cell size in Δ*mreB* cells. The shaded grey areas are indicative of regions where survival significantly drops.

The online version of this article includes the following figure supplement(s) for figure 1:

**Figure supplement 1.** Correlation between cell size and DNA content in SBW25 and Δ*mreB* cells.

**Figure supplement 2.** Cell size and DNA content by flow cytometry.

**Figure supplement 3.** Growth curves for ancestral SBW25 and Δ*mreB* strains as measured by OD$_{600}$ in Lysogeny Broth (LB).

**Figure supplement 4.** Ancestral SBW25 and Δ*mreB* cells, with ectopically added MreB expressed from the *glmS* region of the chromosome.

**Figure supplement 5.** Aspect ratio and orientation of elongation axes.

**Figure supplement 6.** Rotation of elongation axis at division.

# Results

## Deletion of *mreB* from *P. fluorescens* SBW25 generates viable spherical cells

Previous observations of viable *mreB* mutants of SBW25 came from transposon screens of adaptive (wrinkly spreader) mutants (*Spiers et al., 2002*). To establish that cells can indeed proliferate despite the absence of MreB, *mreB* was deleted from ancestral *P. fluorescens* SBW25 (hereafter SBW25). In LB broth, SBW25 Δ*mreB* cells were viable, spherical, and variable in cell volume (*Figure 1A*). DNA sequencing showed no evidence of cryptic compensatory mutations (see Breseq output, https://doi.org/10.17617/3.CU5SX1).

Cell shape, width, and length were used to estimate volume ($V_e$). The volume of Δ*mreB* mutant cells was larger (20.65±16.17 μm$^3$) than those of ancestral SBW25 (3.27±0.94 μm$^3$) cells. A negative correlation between $V_e$ and compactness (the ratio of spherical to circle shape with a compactness of 1.0 being a circle) as a result of cell elongation was evident for SBW25 ($R^2$=0.63) (*Figure 1B*). In contrast, only a weak correlation was observed for the Δ*mreB* mutant ($R^2$=0.13) (*Figure 1B*). As cell volume increases, DNA content is expected to increase provided that DNA replication continues independently of septation. Increased DNA content and spherical cell shape are both predicted to further perturb cell division (*Jun and Mulder, 2006*; *Jun and Wright, 2010*). Ancestral SBW25 and Δ*mreB* cells were stained with a nucleic acid stain (DAPI) and analysed using flow cytometry. In both genotypes DNA content scaled with cell volume. The largest Δ*mreB* cells had many times the DNA content of ancestral cells, scaling approximately with volume (*Figure 1—figure supplement 1*, *Figure 1—figure supplement 2*); small cells yield a signature consistent with maintenance of DNA (*Figure 1—figure supplement 1*).

In comparison with SBW25, the Δ*mreB* mutant had reduced fitness (*Figure 1C*) with a longer effective generation time, extended lag phase, and lower maximum yield (*Figure 1—figure supplement 3*). Complementation of Δ*mreB* with the ancestral gene restored rod-shaped morphology and fitness (*Figure 1—figure supplement 4*).

To understand the causes of reduced fitness, the division of single cells was monitored by time-lapse video microscopy on agar pads. Although SBW25 Δ*mreB* cells appear spherical, measurement of cell aspect ratio revealed existence of an elongation axis that remained constant throughout the cell cycle (*Figure 1—figure supplement 5*). As in rod- and ovococci-shaped bacteria, the septum formed perpendicularly to the elongation axis. After cell division, the elongation axis rotated 90° relative to the mother cell (*Figure 1—figure supplement 6*, *Video 1*), which is reminiscent of ovococci (*Pinho et al., 2013*). But more surprisingly, measurement of the rate of volume extension (hereafter termed elongation rate measured in min$^{-1}$) showed that during exponential phase it was indistinguishable from the cell elongation rate of ancestral SBW25 (*Figure 1D*).

Since elongation rate was unchanged upon deletion of *mreB*, reduced fitness of the mutant was likely due to production of non-viable cells. We thus measured cell viability in LB broth with a 'live/dead' assay and showed that 95% of ancestral SBW25 cells were viable, while the fraction of viable Δ*mreB* mutant cells was reduced to 81%. Thus, the decreased fitness of the mutant is primarily a consequence of the production of non-viable cells (*Figure 1E*). To confirm this result, we performed time-lapse microscopy of cells grown on agar pads to measure the probability p to pass to the next generation. For ancestral SBW25, the probability p of producing offspring equals 1, whereas for Δ*mreB* cells, p=0.85. In instances where the probability to pass to the next generation p is not equal to one, the equation for growth dynamics is represented by $dN = p*r*N*dt$, where $N$ is the number of cells at time $t$ and $r$ is the cell elongation rate. Hence, the effective growth rate of the population is given by p*r. Because Δ*mreB* and ancestral SBW25 cells elongate at the same rate, the fitness of Δ*mreB* cells is directly given by p. Consistently, the probability to pass to the next generation p=0.85 ± 0.05 (*N*=141), as measured in single-cell experiments, is not significantly different from the fitness of 0.77±0.025 for Δ*mreB* in bulk experiments (*Figure 1C*).

We next examined the causes for decreased proliferation capacity by measuring, at the single-cell level, the correlation between the probability p to pass to the next generation and the fidelity of cell division. To this end, the precision of septum positioning, $C_s$, and correlation between the relative difference in elongation rate between sister cells at generation F1, $C_g$, and probability p of producing offspring at the end of generation F1 (*Figure 1F*) were calculated. As a proxy to determine the precision of septum positioning, we measured the relative difference $C_s$ in cell volume $V_e$, between two

daughter cells immediately after division at generation F2. In ancestral SBW25, septum position was accurately placed, with only 5% difference in the volume of sister cells. In ΔmreB mutant cells, the accuracy of septum position was significantly altered, with 19.4% difference in the volume of sister cells. To determine if sister cells had different growth dynamics, we measured the relative difference $C_g$ in elongation rate $r$ between two sister cells at generation F1. In ancestral SBW25, the elongation rate in pairs of sister cells differed by 5%, whereas in ΔmreB cells the difference in rate was on average 55%. Combined together, error in septum positioning and growth asymmetry between sister cells explains why the ΔmreB mutant produces cells of variable volume, which then have different proliferative capacity. To examine how cell volume affects proliferation ability, the probability p to pass to the next generation was plotted as a function of cell volume. As evident in *Figure 1G* the volume distribution has two extreme tails, with both large and small cells being unlikely to produce viable offspring.

## Selection rescues fitness of a compromised *mreB* deletion mutant

To determine whether selection could discover mutation(s) that might compensate for reduced fitness of ΔmreB, SBW25 ΔmreB was propagated for 1000 generations in shaken broth culture with daily transfer. The competitive fitness of derived types was measured relative to ancestral SBW25 using neutral markers to distinguish competitors. Fitness of all replicate lines (relative to ancestral SBW25) significantly improved (*Figure 2A*) to the point where fitness was comparable to ancestral SBW25 (*Figure 2—figure supplement 1*). Subsequent measurement of fitness at generations 50, 250, and 500 showed that compensation of fitness was rapid, with a highly similar trajectory for each of the 10 replicate populations (*Figure 2—figure supplement 2*). Such rapid restoration of fitness is indicative of a small number of readily achievable compensatory mutations.

Examination of derived genotypes at generation 1000 by microscopy showed that all lines retained the coccoid shape (*Figure 2B*, *Figure 2—figure supplement 3*). Cell volume decreased compared to that of the ancestral cell type (*Figure 2B*) and did not overlap with the ΔmreB population (*Figures 1B and 2B*) showing that the derived cells define a new phenotype and not a subset of the spherical ΔmreB ancestor. Analysis of the pattern of cell division in the derived lines showed evidence of an elongation axis, which rotated at cell division, as observed for ΔmreB mutant cells (*Figure 1—figure supplement 6*).

## Identifying compensatory mutations

In order to identify the genetic basis of adaptation, populations were sequenced at generations 500 and 1000 with the ensuing reads mapped to the genome of SBW25 (*Supplementary file 1*) and mutations identified using Breseq (*Deatherage and Barrick, 2014*). A single gene, *pbp1A* (PFLU0406) had mutations in seven independent lines. In four of these, Line 1 (D484N), Line 3 (D721A), Line 4 (T362P), and Line 6 (N698K), the mutation had risen to high frequency, but only in Line 6 was the mutation fixed. Some lines (4, 9, and 10) had more than a single *pbp1A* mutation and in some instances the mutation detected at generation 500 differed from that evident at generation 1000. Clonal interface between competing mutations thus seems likely. In the three lines, with no evidence of *pbp1A* mutations at generations 500, or 1000, two exhibited mutations in genes involved in septum formation (*ftsA* for Line 2 and *ftsZ* for Line 5). Finally, Line 7 contained a five-gene deletion spanning PFLU4921 to PLFU4925 which included *oprD*, but also has a non-synonymous mutation in *ftsE* that was not detected at generation 500, but composed 65% of sequencing reads at generation 1000. The prominence of *pbp1A* mutations led us to further study two lines showing *pbp1A* mutations (Line 1 and Line 4) and to compare them with the five-gene deletion (PFLU4921 to PLFU4925 Δ5399934–5403214) evident in Line 7.

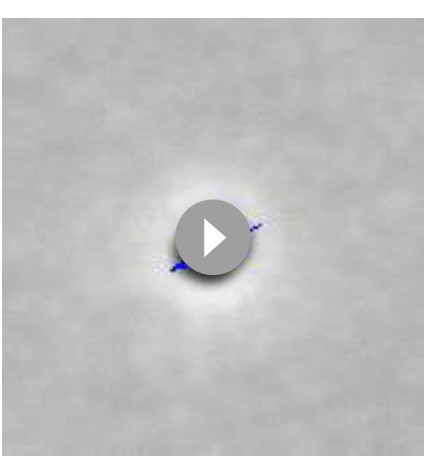

**Video 1.** Elongation axis rotates 90° relative to the mother cell after cell division.

https://elifesciences.org/articles/98218/figures#video1

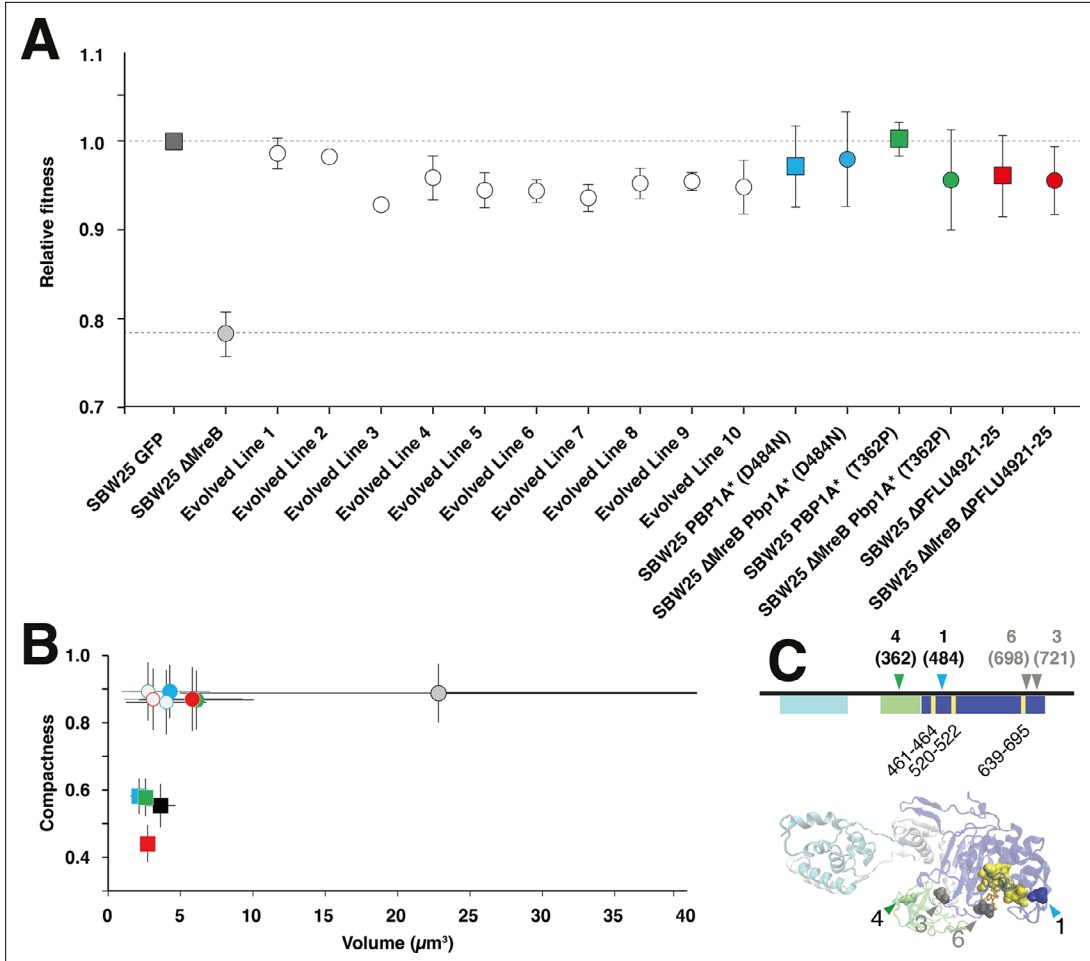

**Figure 2.** Characterisation of evolved lines at 1000 generations. (**A**) Relative fitness of the *ΔmreB* mutant, derived lines after ~1000 generations, and mutant reconstructions in *ΔmreB* and SBW25, relative to SBW25 (dashed line) in pairwise competition experiments. Error bars represent standard deviation. (**B**) Compactness versus estimated volume ($V_e$) of derived lines: Line 1 in blue, Line 4 in green, and Line 7 in red; derived mutations reconstructed in *ΔmreB* and SBW25 backgrounds. Data points are means and SD of X 100 cells. Square data points represent rod-like cells, whereas circles (both open and filled) represent spherical cells. Ancestral genotypes are labelled in black. Open squares and open circles depict the reconstruction of mutations in the SBW25 and *ΔmreB* backgrounds, respectively. Filled circles represent the 1000 generation evolved lines from where the mutations have been identified. Colours as follows: blue is Line 1 or Pbp1A D484N, green is Line 4 or Pbp1A T362P, red is Line 7. (**C**) Domain map and model of Pbp1A (PFLU0406) showing the two major domains: the glycosyltransferase (GT) domain (cyan) and the transpeptidase (TPase) domain (blue). The oligonucleotide/oligosaccharide binding (OB) domain is shown in green. The active site of the TPase domain is also shown (yellow). The locations of mutations from Lines 1 and 4 are indicated with coloured arrows in the *pbp1A* gene map and in the model of Pbp1A. The mutations in Lines 3 and 6 are also shown in grey but were not characterised further in this work.

The online version of this article includes the following figure supplement(s) for figure 2:

**Figure supplement 1.** Growth curves of evolved *ΔmreB* lineages after 1000 generations of evolution.

**Figure supplement 2.** Relative fitness of the evolved lines at distinct points during the 1000 generations of evolution.

**Figure supplement 3.** Phase contrast images of the 10 independent evolved lines.

**Figure supplement 4.** A Clustal Omega sequence alignment of a segment of Pbp1A amino acid sequences from seven bacterial species, including *Acinetobacter baumannii*, which was used for the construction of the protein model showing the location of the evolved line mutations.

**Figure supplement 5.** Growth curves of the reconstructions in (**A**) SBW25 and (**B**) *ΔmreB* backgrounds.

**Figure supplement 6.** Elongation rate measures the rate, *r*, at which cell volume increases $V(t)=V_0 e^{rt}$.

**Figure supplement 7.** Fitness of SBW25 *ΔmreB Δpbp1A* and *ΔmreB pbp1A* (T362P) relative to SBW25 *ΔmreB*.

**Figure supplement 8.** Cell morphologies.

**Figure supplement 9.** The cell morphologies and relationship between cell shape and estimated volume ($V_e$) represented using compactness as in *Figure 1B* of mutation reconstructions.

Figure 2 continued

**Figure supplement 10.** Genome map of the *oprD*-inclusive deletion (PFLU4921–4925 Δ5399934–5403214) and surrounding region.

**Figure supplement 11.** Reconstruction of Line 1, 4, and 7 mutations in ancestral SBW25 (WT) and SBW25 Δ*mreB* background stained with DAPI.

The *pbp1A* gene is a class A PBP with both TPase and TGase activities. It is semi-autonomous and interacts with the rod complex and specifically with PBP2 (*Banzhaf et al., 2012*; *Geisinger et al., 2020*). Pbp1A contains three known domains (*Figure 2C*). Structure mapping of the mutations shows that the mutation in Line 1 occurred in the TPase domain, which is proximal to the active site (*Figure 2—figure supplement 4*). Similar mutations in *Streptococcus pneumoniae* cause a loss-of-function (*Job et al., 2008*) and rescue viability of both *mreC* and *mreD* mutants (*Land and Winkler, 2011*). The mutation in Line 4 occurred in the oligonucleotide/oligosaccharide binding or outer membrane Pbp1A docking domain (OB/ODD) (*Typas et al., 2011*). Each mutation was reconstructed by allelic exchange in SBW25 and SBW25 Δ*mreB* backgrounds.

Given the very rapid fitness increase (*Figure 2—figure supplement 2*), we looked for evidence of *pbp1A* mutations at generation 50. The gene was amplified by PCR from 16 colonies from each replicate population and the product Sanger sequenced. All lines had at least one mutation (out of the 16 colonies), demonstrating the relative ease at which *pbp1A* mutations arise. Results are shown in *Supplementary file 2*.

## Analysis of *pbp1A* mutations

Both Δ*mreB pbp1A* reconstructed genotypes (hereafter Δ*mreB pbp1A\**) produced spherical-to-ovoid cells with near ancestral SBW25 volume (Δ*mreB pbp1A* (D484N) $M$=3.86 ± 0.89 $\mu m^3$; Δ*mreB pbp1A* (T362P) $M$=5.51 ± 0.87 $\mu m^3$) and DNA content (*Figure 2B*, *Figure 1—figure supplement 2*). Analysis of the pattern of cell division in the reconstructed lines showed evidence of an elongation axis that rotated at cell division as observed for SBW25 Δ*mreB* (*Figure 1—figure supplement 6*). Δ*mreB pbp1A\** genotypes showed growth dynamics and fitness effects similar to ancestral SBW25 (*Figure 2A*, *Figure 2—figure supplement 5*), indicating that these *pbp1A* mutations are each sufficient to restore ancestral fitness and to recapitulate the major phenotypes of derived Lines 1 and 4. Elongation rate was identical to SBW25 for SBW25 Δ*mreB* carrying the reconstructed mutations (*Figure 2—figure supplement 6*). Fitness of a genetically engineered Δ*mreB* Δ*pbp1A* mutant (relative to Δ*mreB*) was 1.35 (SD±0.04, *n*=6) was indistinguishable from fitness of the Line 4 mutation reconstructed in the same Δ*mreB* background (1.38; SD±0.11, *n*=6), demonstrating that loss-of-function of *pbpB1A* is sufficient to restore fitness (*Figure 2—figure supplement 7*). Loss-of-function is also consistent with the finding that at generation 50 all lines harboured mutations in *pbp1A* (*Supplementary file 2*).

When reconstructed in ancestral SBW25, *pbp1A* mutations had little effect on population dynamics (*Figure 2—figure supplement 5*) or shape (*Figure 2B*), however, elongation rates were reduced (*Figure 2—figure supplement 6*), suggesting that the *pbp1A* mutations result in a reduction – or more likely even complete loss-of-function. In addition, cells were narrower than WT cells: 0.89 μm (SD±0.07) and 0.94 μm (SD±0.05) for Pbp1A TPase and OB/ODD mutations, respectively, compared to 1.00 μm (SD±0.06) for ancestral SBW25. This resulted in smaller cell volumes (*Figure 2B*, *Figure 2—figure supplement 8*, and *Figure 2—figure supplement 9*). The reduced widths/volumes – which also correlate with reduced DNA content (*Figure 1—figure supplement 2*) – might be indicative of reduced enzymatic function leading to reduced lateral cell wall synthesis as the case for loss-of-function mutations in *pbp1A* reported in both *B. subtilis* and *E. coli* (*Murray et al., 1998*; *Claessen et al., 2008*; *Kawai et al., 2009*; *Banzhaf et al., 2012*).

## Analysis of Δ*pflu4921–4925*

The five-gene deletion (Δ5399934–5403214 (Δ*pflu4921–4925*)) in derived Line 7 (*Figure 2—figure supplement 10*, *Supplementary file 1*) includes three hypothetical proteins, a cold shock protein (PFLU4922, encoding CspC), and an outer membrane porin, PFLU4925, which encodes OprD. The latter is responsible for the influx of basic amino acids and some antibiotics into the bacterial cell (*Skurnik et al., 2013*). The deletion was also reconstructed and characterised in the Δ*mreB* and ancestral SBW25 backgrounds. The relative fitness of the Δ*mreB* Δ*pflu4921–4925* genotype was similar to the ancestral SBW25 type (*Figure 2A*), but with an extended lag time (*Figure 2—figure supplement*

5). Cells were spherical with an average $V_e$ of 5.26 μm³ (SD±3.13) (*Figure 2B*, *Figure 2—figure supplement 8*, and *Figure 2—figure supplement 9*) and harboured less DNA compared to Δ*mreB* SBW25, but more than the Pbp1A* reconstructions in the same background (*Figure 1—figure supplement 2*). In addition, a fraction of the cell division events were defective, resulting in clumps of spherical cells with imperfectly formed septa (*Figure 2B*, *Figure 2—figure supplement 8*, *Figure 2—figure supplement 9*, *Figure 2—figure supplement 10*, and *Figure 2—figure supplement 11*). Spherical cells with notably incomplete septa were not observed in the derived Line 7 population (*Supplementary file 1*).

When genes *pflu4921–4925* (Δ5399934–5403214) were deleted from ancestral SBW25 the cells remained rod-shaped and displayed aberrant growth characteristics relative to the ancestral strain (*Figure 2—figure supplement 8*). The elongation rate was slower (*Figure 2—figure supplement 6*), cells were significantly thinner (width = 0.74 μm SD±0.06; p<0.001, two-sample *t*-test), and had a smaller average $V_e$ (2.47 μm³, SD±1.18) (*Figure 2B* and *Figure 2—figure supplement 9*). As in the Δ*mreB* background, a fraction of cells failed to completely separate at cell division and DNA was dispersed between incomplete septa (*Figure 2—figure supplement 8*, *Figure 2—figure supplement 9*, *Figure 2—figure supplement 10*, and *Figure 2—figure supplement 11*). The average DNA content of SBW25 Δ*pflu4921–4925* cells was greater than the *pbp1A** reconstructions in the same background (*Figure 1—figure supplement 2*).

## Cell volume homeostasis

The large average cell volume observed in stationary phase SBW25 Δ*mreB* cells (*Figure 2B*) suggests that some of the physical parameters governing cell volume may have been altered. Cell volume at steady state is determined by the equilibrium between turgor pressure (*Rojas and Huang, 2018*) and membrane tension through the Laplace law for thin elastic shells (*Le Verge-Serandour and Turlier, 2021*) (see Materials and methods for elaboration). Since it is difficult to imagine a scenario where the cell wall would strengthen in the absence of *mreB*, *mreB* loss is expected to result in a significant decrease of internal pressure compared to ancestral SBW25 cells, synonymous with the influx of water through pores formed in the cell wall.

During exponential growth, the cell volume of daughter cells is limited by the amount of new material that can be incorporated by the mother cell. According to the 'adder' model (*Campos et al., 2014*; *Taheri-Araghi et al., 2017*), the average cell volume in the population converges within a few generations towards the average added volume. We thus quantified the added volume for all strains (*Figure 3A*) and confirmed that cell volume rapidly reduced for exponentially growing SBW25 Δ*mreB* cells (*Video 2*). Since the volume of Δ*mreB* cells in exponential phase is much lower than in stationary phase, it suggests that cells swell during stationary phase, probably due to defects in cell wall integrity with deleterious consequences for cell viability. Hence, without restoring cell wall integrity, cell volume homeostasis cannot be maintained. Remarkably, the added volumes of spherical cells in each of the derived mutant lines (*Figure 3A*) were close to the volumes corresponding to the passage from a cylindrical to spherical geometry with constant turgor pressure and membrane tension (see Materials and methods). This suggests that adaptive mutations restored the mechanical integrity of the cell wall.

To maintain cell volume homeostasis, elongation and septation rates must be balanced so that cells maintain a steady volume throughout successive generations. Indeed, if septation is faster than elongation, cell volume of progeny decreases, while if elongation is faster than the rate of septation rate, cells increase in volume. In both instances the fate of populations is extinction. The rate of septation was significantly higher than the elongation rate in Δ*mreB* cells showing that the distribution of cell volume is unstable and confirms that the average volume is expected to decrease in subsequent generations (*Figure 3B*). In contrast to SBW25 Δ*mreB* cells, the rates of elongation and septation were balanced in the derived lines (*Figure 3B*), indicating that cell volume homeostasis has been reached. Reconstruction of *pbp1A* mutations in SBW25 Δ*mreB* also restored this balance, suggesting that the mutations contribute to maintenance of volume homeostasis in the absence of *mreB*. In contrast, the Line 7 deletion reconstructed in SBW25 Δ*mreB* did not restore this balance (*Figure 3B*).

Another important feature for cell volume homeostasis is the fidelity of cell division: two daughter cells with equal volume must themselves generate four daughters of equal volume. Therefore, we measured the relative difference in elongation rates ($C_g$) and division time ($C_T$) between sister cells at generation F1 on agar pads, as well as the relative difference in precision of septum positioning ($C_s$) for all derived and reconstructed lines. All evolved lines regained symmetry both in elongation rate

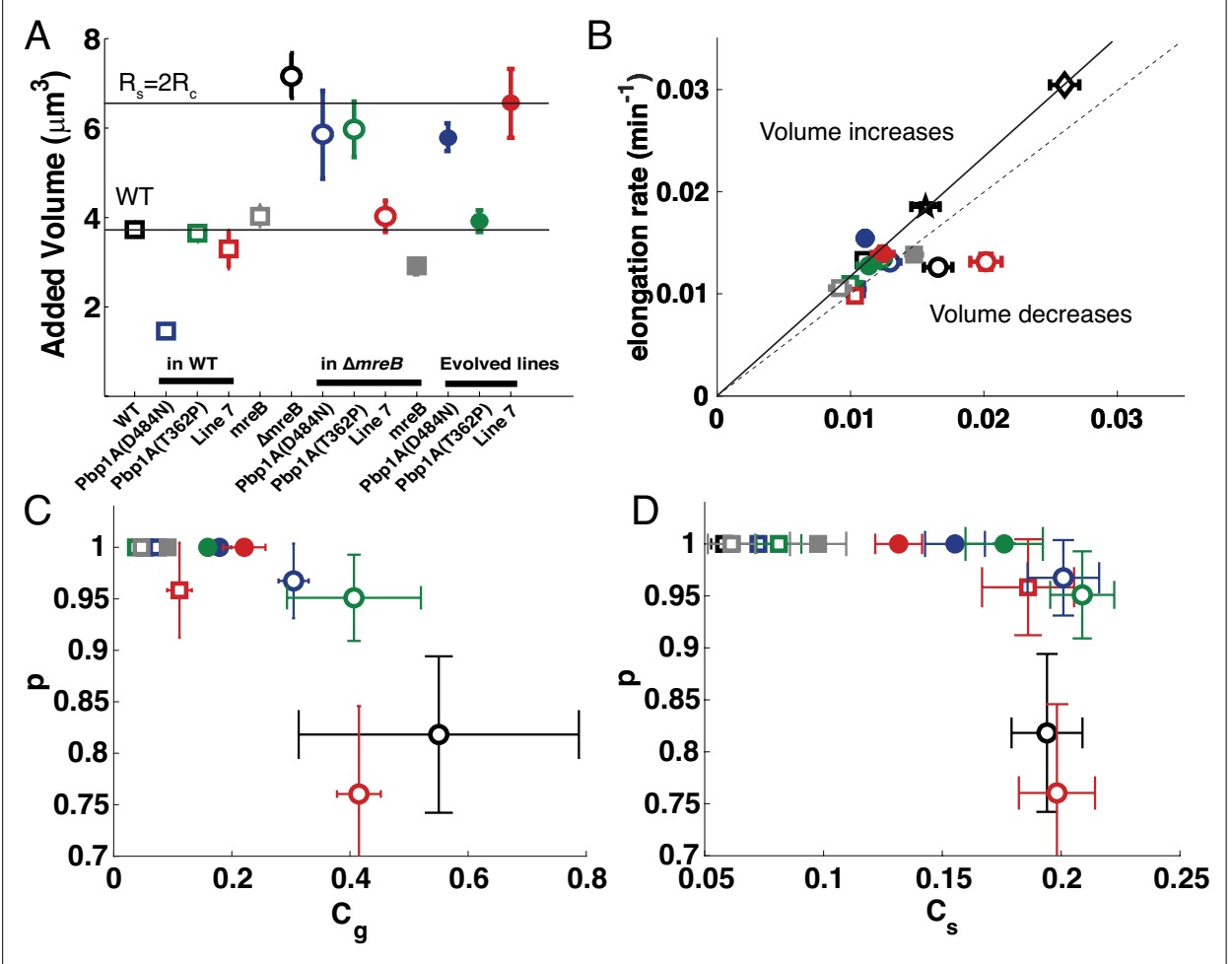

**Figure 3.** Cell size homeostasis. (**A**) Average added volume for the different genotypes, which is cell volume at septation less cell volume at birth. The horizontal lines correspond to the volume of ancestral SBW25 cells and to their spherical counterparts ($R_s = 2*R_c$). (**B**) Elongation rate plotted against septation rate. The black line corresponds to size homeostasis of wild-type strains (square = *P. fluorescens*; star = *Pseudomonas aeruginosa*; diamond = *E. coli*). Below this line, septation is faster than elongation and cell size decreases. The dashed line corresponds to equal septation and elongation rates. (**C**) The probability p to pass to the next generation as a function of growth asymmetry $C_g$. (**D**) The probability p to pass to the next generation as a function of error in septum positioning measured as the relative cell size between daughters at division $C_s$. In (**A, B, C, and D**) square data points (both open and filled) represent rod-like cells, whereas circles (both open and filled) represent spherical cells. Ancestral genotypes are labelled in black. Open squares and open circles depict the reconstruction of mutations in the WT and Δ*mreB* backgrounds, respectively. Filled circles represent the evolved lines where the mutations have been identified. Colours as in *Figure 2*; blue is Line 1 or Pbp1A D484N, green is Line 4 or Pbp1a T362P, red is Line 7, grey is ectopic *mreB* expression. Error bars represent standard errors. In (**A, B, and D**): ($N_{WT} = 94$; $N_{L1\_WT} = 52$; $N_{L4\_WT} = 38$; $N_{L7\_WT} = 72$; $N_{mreB\_WT} = 36$; $N_{\Delta mreB} = 99$; $N_{L1\_\Delta mreB} = 92$; $N_{L4\_\Delta mreB} = 102$; $N_{L7\_\Delta mreB} = 96$; $N_{mreB\_\Delta mreB} = 64$; $N_{L1}=88$; $N_{L4}=74$; $N_{L7}=76$); in (**C**) ($N_{WT} = 47$; $N_{L1\_WT} = 26$; $N_{L4\_WT} = 19$; $N_{L7\_WT} = 38$; $N_{mreB\_WT} = 18$; $N_{\Delta mreB} = 55$; $N_{L1\_\Delta mreB} = 56$; $N_{L4\_\Delta mreB} = 62$; $N_{L7\_\Delta mreB} = 53$; $N_{mreB\_\Delta mreB} = 32$; $N_{L1}=44$; $N_{L4}=42$; $N_{L7}=40$).

The online version of this article includes the following figure supplement(s) for figure 3:

**Figure supplement 1.** The probability p to pass to the next generation as a function of division time asymmetry $C_T$.

and in precision of septum positioning, but was most notable for the symmetry of elongation rate (*Figure 3C and D*). These improvements correlate with an increase in proliferative capacity, while no significant correlation was observed in the relative difference of division time $C_T$ between sister cells (*Figure 3—figure supplement 1*). Reconstructions in ancestral SBW25 showed a slight alteration of symmetry, which is more pronounced in the case of Line 7 (five-gene deletion). Reconstructions in evolved lines Line 1 and Line 4 showed improvement in growth asymmetry. Interestingly, recovery was only partial for Line 7, raising the possibility of additional effects due to a secondary mutation whose effects depend on the mechanics of the environment (liquid versus solid culture). Indeed, as indicated above and shown in *Supplementary file 1*, Line 7, at generation 1000, also contains a

non-synonymous mutation in *ftsE* (T188A) that had reached high frequency. Given that FtsE promotes elevated peptidoglycan synthesis at cell septa the possibility of synergistically beneficial effects with the OprD-inclusive deletion is not improbable (*Mallik et al., 2023*).

The mutations observed in Line 1 or Line 4 showed a significant improvement in the precision of septum position, suggesting that asymmetry in elongation rate between daughter cells has greater impacts on cell survival than septum positioning. This can also be seen from the shape of the curves (*Figure 3C and D*), where p decreases linearly with growth asymmetry up to 50%, while precision of septum positioning saturates at 20% regardless of p, probably because nucleoid occlusion is able to achieve a sufficiently good level of septum positioning.

## Investigating the molecular consequences of mutations

To obtain insight into the molecular mechanisms by which mutations in *pbp1A* compensate for loss of *mreB*, we performed a range of biochemical analyses on SBW25 and SBW25 Δ*mreB* engineered with the fitness-rescuing mutations in *pbp1A*: SBW25/SBW25 Δ*mreB* Pbp1A (D484N) (mutation in the TPase domain); SBW25/SBW25 Δ*mreB* Pbp1A (T362P) (mutation in the OB/ODD). SBW25 Δ*mreB* Δ*pbp1A* was used as a negative control. In ancestral SBW25 36.34% (SD±2.26) of muropeptides are cross-linked (*Figure 4A*). The Δ*mreB* mutant showed a marked increase in cross-linking (40.47% SD±1.50). The degree of cross-linking in the *pbp1A* mutants in either SBW25 or the Δ*mreB* genetic background were lower (31.28% SD±0.98 to 33.66% SD±0.52).

Lower levels of cross-linking observed in the Δ*mreB* and SBW25 reconstructed lines suggests that mutations in Lines 1 and 4 either decreased the amount of transpeptidation or increased the amount of transglycosylation. To discriminate between the two hypotheses, we utilised Bocillin-FL labelling, a fluorescent penicillin derivative that binds to the active site of PBPs, as a proxy to measure PBP1a TPase activity. We included SBW25 Δ*pbp1A* as a control to specifically identify the band corresponding to Pbp1A in the SDS-PAGE gel (*Figure 4B*). In agreement with higher levels of cross-linking activity measured in SBW25 Δ*mreB* (*Figure 4A*), the intensity of the Pbp1A band was slightly higher in Δ*mreB* cells compared to the ancestral genotype. In comparison to the ancestor of the evolved lines (SBW25 Δ*mreB*), this signal was little changed in the Line 1 mutant (TP domain (SBW25 Pbp1A (D484N))) (*Figure 4B*), but absent in the Line 4 mutant (OB/ODD (SBW25 Pbp1A (T362P))). The activity of the OB/ODD is not well characterised, but is thought to be an important regulatory domain that licences TPase domain function through interaction with LpoA (*Sardis et al., 2021*). The relative peptidoglycan density and UPLC traces are shown (*Figure 4—figure supplement 1*).

In addition, we studied how the Δ*mreB* and *pbp1A* OB/ODD and TPase domain mutations affect the location of active cell wall synthesis. This was achieved by incorporation of fluorescently labelled D-amino acids (FDAAs) via brief pulses of BADA into exponentially growing cells followed by microscopy to visualise locations of active cell wall synthesis (*Figure 4C* and *Figure 4—figure supplement 3*). In SBW25, BADA incorporation into the cell wall was homogeneous across the lateral surface during growth, but enriched at the mid-cell during septation. In SBW25 Δ*mreB* cells, BADA labelling was heterogeneous across the entire cell wall. This is contrary to expectation – at odds with observations of diffuse and homogeneous wall synthesis in MreB-depleted *E. coli* (*Ursell et al., 2014*) – and indicates that even in the absence of MreB, cell wall synthesis is active. SBW25 Δ*mreB* containing *pbp1A\** mutations resulted in small spherical cells in which the prominence of cell wall synthesis in the septal region was restored as the primary location of peptidoglycan production, particularly in longer cells (*Figure 4C*). Thus, *pbp1A* mutations appear to restore cell homogeneity by eliminating disorganised patterns of cell wall synthesis (*Figure 4C*).

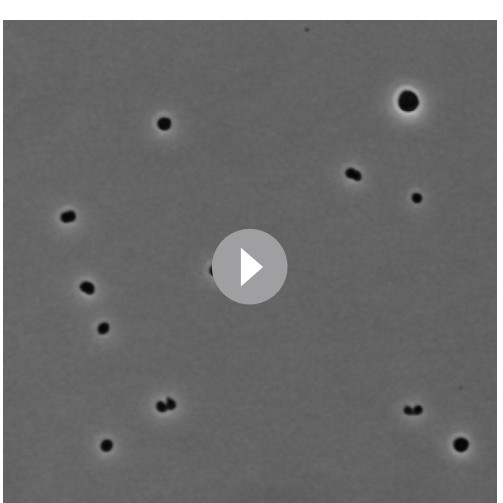

**Video 2.** Reduction in cell volume for exponentially growing SBW25 Δ*mreB* cells.
https://elifesciences.org/articles/98218/figures#video2

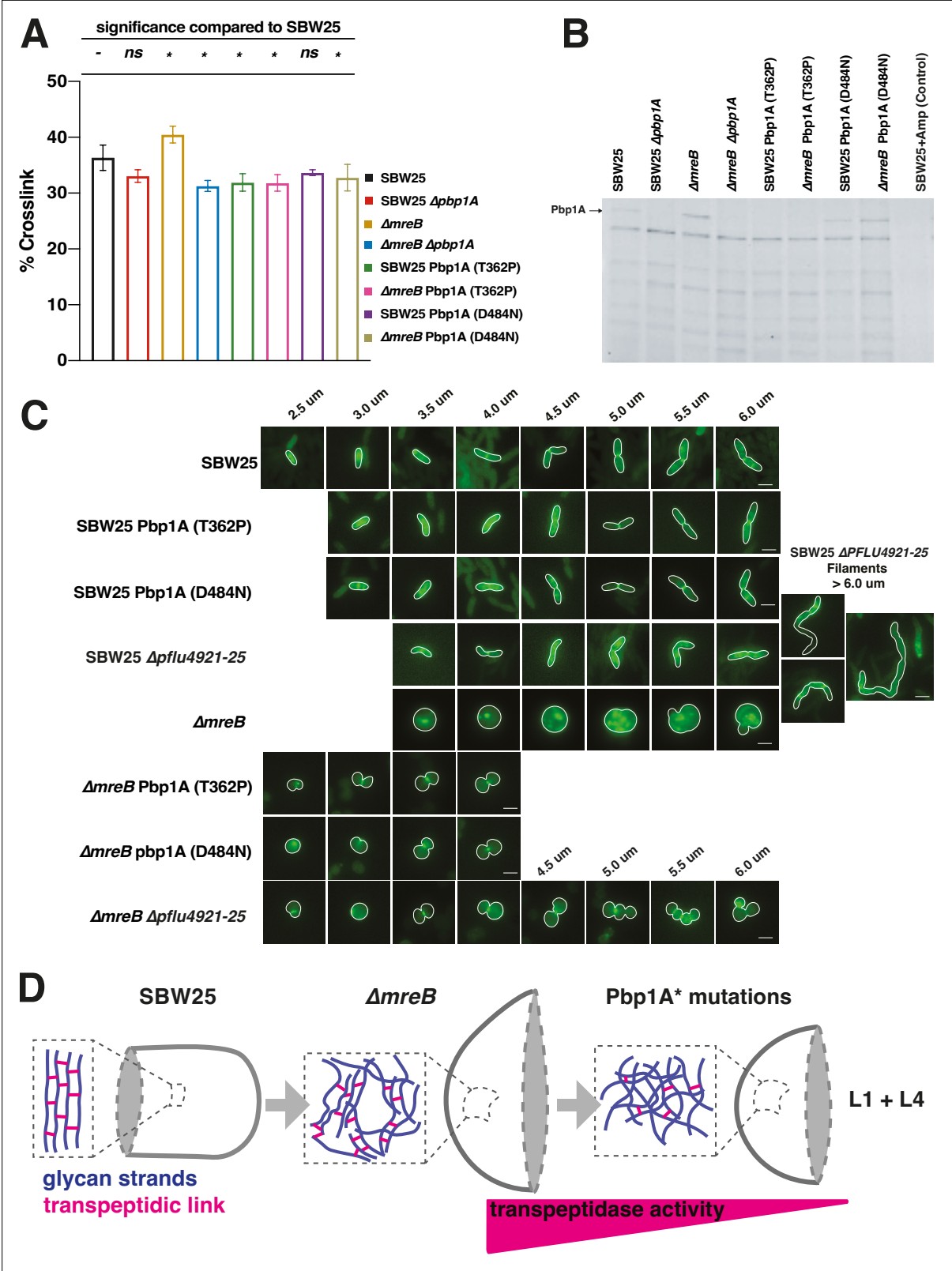

**Figure 4.** Molecular investigation of adaptive mutations in Δ*mreB* evolution. (**A**) Peptidoglycan cross-linking in the cell walls of mutants in SBW25 and Δ*mreB* genetic backgrounds. The error bars (SD) are the result of three biological replicates. ANOVA revealed a highly significant difference among means [$F_{7,16}$ = 8.19, p<0.001] with Dunnett's post hoc test adjusted for multiple comparisons showing that five genotypes (*) differ significantly (p<0.05) from SBW25. (**B**) Cell extracts subjected to polyacrylamide gel and SDS-PAGE labelling with Bocillin-FL gel to demonstrate penicillin binding of Pbp1A

*Figure 4 continued on next page*

*Figure 4 continued*

(labelled) and other Pbps in *P. fluorescens* SBW25. See *Figure 4—figure supplement 2* for quantification of band intensities and *Figure 4—source data 1* for original gel image. (**C**) Fluorescent images of mutations in SBW25 and Δ*mreB* backgrounds subjected to short-pulse labelling with fluorescent D-amino acid BADA to reveal the location and organisation of cell wall construction in the mutant backgrounds. As cells elongate in preparation to divide, septal peptidoglycan synthesis becomes more pronounced. Δ*mreB* cells have highly disorganised synthesis (see *Figure 4—figure supplement 3* for quantitative analysis), which is alleviated in Δ*mreB* Pbp1A* and Δ*mreB* ΔPFLU4921-25 mutants. Cell outlines from phase contrast images show the cell boundaries. Scale bars = 2 µm. (**D**) Model of evolutionary rescue in Δ*mreB* mutant cells highlighting the organisation of the predicted glycan meshwork in the rod and spherical geometries. In Pbp1A* mutants (Line 1 and Line 4), a decrease in transpeptidase activity favours disorganised cell wall architecture, which is better suited to the topology of a sphere.

The online version of this article includes the following source data and figure supplement(s) for figure 4:

**Source data 1.** Original gel image shown in *Figure 4B*.

**Source data 2.** Labelled version of *Figure 4—source data 1*.

**Figure supplement 1.** Peptidoglycan analysis.

**Figure supplement 2.** Quantification of band intensity corresponding to PBP1A TPase activity as presented in the gel shown in *Figure 4B*.

**Figure supplement 3.** Quantification of heterogeneity in cell wall synthesis as determined by BADA labelling (see *Figure 4C*).

## Discussion

The ability to generate a viable *mreB* deletion mutant of *P. fluorescens* SBW25 – albeit with significantly compromised fitness – has provided opportunity to investigate the phenotypic consequences of loss of MreB function without use of MreB-inhibiting drugs and to explore mutational routes leading to fitness restoration.

After 250 generations of selection, fitness of all derived lines was restored to wild-type levels (*Figure 2A*). Measurement of fitness at earlier time points showed that compensation occurred early in the selection regime and in single steps (*Figure 2A*, *Figure 2—figure supplement 2*), consistent with compensation occurring via one or a small number of mutations. Sequencing and genetic reconstruction experiments confirmed, particularly for the class of mutant with defects in *pbp1A*, that single point mutations reducing or eliminating gene function were sufficient. In all derived lines, cell shape remained spherical. This sits in accord with evidence suggesting that there are no genetic routes or growth conditions able to restore rod-like shape in the absence of MreB (*Shi et al., 2018*). This stated, we recognise that rod-shaped bacteria belonging to the *Actinobacteria* and *Rhizobiales* lack MreB (*Margolin, 2009*; *Zhao et al., 2021*), suggesting that in principle it is possible to restore rod-shape in the absence of MreB.

In experiments where selection is used to find mutations that compensate for deleterious effects arising from deletion of a gene that compromises fitness (*Laan et al., 2015*; *Lind et al., 2015*; *Rainey et al., 2017*), a central issue concerns the focus of selection: the defect that selection might stand to correct. In this work, where deletion of *mreB* provided the compromised starting genotype, the most obvious difference between SBW25 Δ*mreB* and each of the 10 derived lines was cell volume. It is therefore tempting to suggest that volume is a target of selection, and especially so given known adaptive changes in cell volume in the long-term Lenski experiment (*Lenski and Travisano, 1994*; *Monds et al., 2014*). However, a closer examination of data, particularly from time-resolved microscopy, patterns of peptidoglycan synthesis, and biophysical considerations, suggests a more complex set of effects and adaptive responses.

At first glance, the most notable distinction between cells of ancestral and derived types was shape and volume: SBW25 cells are rod-shaped (thus with low compactness) with relatively little variance in the volume of daughter cells, whereas the shape of Δ*mreB* cells tends towards spherical with large variance in volume. While the rate of elongation is the same in SBW25 and Δ*mreB* cells, substantial variance was apparent in the rate of elongation among daughter (Δ*mreB*) cells (*Figure 1D*). Also evident were significant differences in cell viability, with viability of daughter cells being high in SBW25, but much reduced in the Δ*mreB* mutant. With focus on Δ*mreB* cells, viability of daughter cells depends on cell volume and declines significantly in both very large and very small cells (*Figure 1G*). This is understandable in terms of effects on DNA segregation, with small cells proving non-viable most likely through lack of DNA, and large cells being similarly non-viable as a consequence of multiple copies of intertwined chromosomes and imbalance of osmotic pressure.

A key issue concerns the causes of variability in cell volume, towards which both septation asymmetry ($C_s$) and differential elongation rate ($C_g$) between daughter cells are the primary contributory factors. As evident by comparing $C_s$ and $C_g$ for ancestral SBW25 (black open square in *Figure 3C and D*) and Δ*mreB* (blue open circle in *Figure 3C and D*) cells, respectively, the greatest difference is the relative difference in elongation rate among sister cells ($C_g$). This indicates that this is the primary cause of variability in cell volume in the *mreB* mutant. Moreover, if correct, it leads to the prediction that fitness restoration will depend on selection finding mutational routes to restore population-level homogeneity in elongation rate among offspring.

What then underlies differences in elongation rate? Here, data on peptidoglycan cross-linking, TPase activity, and patterns of cell wall synthesis in SBW25 versus Δ*mreB* cells are suggestive. Relative to SBW25, cells of the *mreB* mutant showed increased peptidoglycan cross-linking (*Figure 4A*), likely resulting from enhanced TPase activity of Pbp1A (*Figure 4B*) and heterogeneous (uneven) patterns of cell wall synthesis (*Figure 4C*). These data indicate that despite the absence of MreB, cell wall synthesis continues but is dysregulated in space and time. This is in marked contrast to cell wall synthesis in MreB-impaired (by treatment with A22) *E. coli* where cell wall synthesis is diffuse and homogeneous (*Ursell et al., 2014*).

Just what contributes to cell wall synthesis on removal of MreB is unclear. The standard view is that on deletion (or depolymerisation) of MreB, the elongasome is registered non-functional (*Park and Uehara, 2008*), but compelling evidence appears to be lacking (*Cho et al., 2016*). In *E. coli*, PBP2 can trigger elongasome initiation independently of MreB by promoting interaction with a hypothetical protein that in turn interacts with PBP2 and the cell wall (*Özbaykal et al., 2020*). Additionally, in *A. baumannii* high Pbp1A activity inhibits elongasome function (*Simpson et al., 2021*) with Δ*ponA* (*pbp1A*) mutants having enhanced elongasome function (*Kang et al., 2021*). Accordingly, in the following discussion we present two hypotheses to account for our findings: the first assumes that the elongasome is non-functional, whereas the second considers the possibility that the elongasome remains functional, but is stalled (or trapped).

Irrespective of whether the elongasome is rendered non-functional on removal of MreB, it is necessary to explain the observed heterogeneity of cell wall synthesis. Assuming, firstly, a non-functional elongasome, then obvious candidates for cell wall synthesis are the aPBPs, Pbp1A, and Pbp1B, that can function independently of the elongasome (*Cho et al., 2016*) (although depend on activation by LpoA and LpoB, respectively), with Pbp1B being particularly important in repair of cell wall lesions (*Vigouroux et al., 2020*). Perhaps heterogeneity of cell wall synthesis in Δ*mreB* arises from attempts by Pbp1B to repair large-scale lesions, but there being insufficient bound (active) Pbp1B to fully accomplish this.

Assuming that the peptidoglycan-synthesising activity of the elongasome remains active on deletion of *mreB*, then heterogeneity of cell wall synthesis indicates that the multi-protein complex has become either stalled or trapped. This is reminiscent of observations in *B. subtilis* where inactivation of MreB causes failure in the dynamic redistribution of PBP1 (*Kawai et al., 2009*). Stalling – and concomitant heterogeneity of synthesis – could arise from elevated TPase activity that could in turn stem from altered interactions between Pbp2 and Pbp1A, which are known to mutually affect the catalytic activity of the other (*Banzhaf et al., 2012*; *Egan et al., 2020*). In support of this hypothesis, the TPase activity of Pbp1A in SBW25 Δ*mreB* is elevated compared to the same protein in ancestral SBW25 (*Figure 4B*).

In the populations derived from the 1000-generation selection experiment, solutions to variability in cell volume stemmed largely from reduction (or complete abolition) of Pbp1A function. As predicted above and shown in *Figure 3D*, derived lines showed greatest improvement in the degree to which daughter cells differ in elongation rate ($C_g$). Such improvement correlates directly with significant increases in the probability that daughter cells are viable. By way of comparison changes in the relative positioning of the septum among daughter cells in derived lines ($C_s$) were relatively minor. Selection thus appears to have restored fitness by changes that reduced variation in the rate of elongation among daughter cells. Assays of TPase and peptidoglycan cross-linking indicate that this is linked to reduction of TPase function of Pbp1A and a reduction in peptidoglycan cross-linking, with the two together likely favouring a disorganised cell wall architecture suited to the topology of spherical cells (*Figure 4D*).

That selection compensated for the maladapted effects of *mreB* deletion by mutations in *pbp1A* can now be viewed in light of the two hypotheses for heterogeneity in cell wall synthesis. In the case of heterogeneity stemming from absence of elongasome function and localised activity of Pbp1B – and drawing upon the findings that inactivation of Pbp1A in *E. coli* results in a fourfold elevation of the amount of bound (and thus active) Pbp1B (*Vigouroux et al., 2020*) – inactivation of Pbp1A is expected to significantly increase Pbp1B activity perhaps to a level that would allow for wall repair uniformly throughout the cell (*Vigouroux et al., 2020*). Should heterogeneity in cell wall synthesis arise from a stalled, but otherwise functional elongasome, then elimination of Pbp1A might serve to correct the balance between TPase and GTase activity and thus allow the elongasome to move more diffusively along the inner membrane (*Simpson et al., 2021*). Future work is planned to test these competing hypotheses aided by the fact that deletion of *mreB* in SBW25 is not lethal.

The reason that compromised, albeit viable Δ*mreB* cells, can be generated in SBW25 under standard laboratory conditions is unknown, but answers may shed new light on the complexities of elongasome function and particularly the connection between elongasome function and MreB. Deletion of *mreB* in many bacteria is lethal including the closely related *P. aeruginosa* (*Robertson et al., 2007*), although in *E. coli* the defect can be rescued by slow growth or by increased activity of the cell division protein FtsZ (*Kruse et al., 2005*; *Bendezú and de Boer, 2008*). In *B. subtilis* lethality can be ameliorated by growth in the presence of magnesium (*Formstone and Errington, 2005*). This suggests that factors contributing to lethality involve subtleties in interactions among elongasome components (including associated factors such as *pbp1A* and *pbp1B*), which are dysregulated in SBW25 Δ*mreB*, but are insufficiently out of kilter to result in a lethal phenotype.

In this context there exists possible explanations for the repeated and independent transitions from ancestral rod-shape cells to cocci (*Siefert't and Fox, 1998*). Recent work on the rod to coccus transition in nasopharyngeal pathogens of the Neisseriaceae family, where deletion of *mreB* and other determinants of the Rod complex, while not lethal, result in a significant fitness decrease (*Veyrier et al., 2015*). Such reductions in fitness were not apparent when coccoid cells arose from deletion of the FtsZ ring assembly-promoting protein YacF (ZapD), suggesting the requirement for a series of events beginning with loss of *yacF* followed by evolutionary compensation with eventual loss of the elongasome machinery (*Veyrier et al., 2015*). Our work here shows that the evolutionary route from rod to sphere is readily achievable under laboratory conditions via loss of *mreB* followed by compensatory mutation in *pbp1A* and reorganisation of peptidoglycan structure. Whether such a route is ever realised in nature will depend on the initial fitness effects of the *mreB* mutation, which in turn will depend on environmental context, but it is not implausible that inactivation of *mreB* may confer a fitness advantage under some conditions. This argument is further elaborated by *Yulo and Hendrickson, 2019*.

Given the complexity of the cell wall synthesis machinery that defines rod-shape in bacteria, it is hard to imagine how rods could have evolved prior to cocci. However, the cylindrical shape offers a number of advantages. For a given biomass (or cell volume), shape determines surface area of the cell envelope, which is the smallest surface area associated with the spherical shape. As shape sets the surface/volume ratio, it also determines the ratio between supply (proportional to the surface) and demand (proportional to cell volume). From this point of view, it is more efficient to be cylindrical (*Young, 2006*). This also holds for surface attachment and biofilm formation (*Young, 2006*). But above all, for growing cells, the ratio between supply and demand is constant in rod-shaped bacteria, whereas it decreases for cocci. This requires that spherical cells evolve complex regulatory networks capable of maintaining the correct concentration of cellular proteins despite changes in surface/volume ratio. From this point of view, rod-shaped bacteria offer opportunities to develop unsophisticated regulatory networks.

While fitness in the majority of experimental lines was restored by mutations in *pbp1A*, one line, Line 5, contained a mutation in *ftsZ*, but oddly this mutation, while apparent at generation 500, was undetected by population sequencing at generation 1000 (*Supplementary file 1*). Interestingly, the FtsZ D176A mutation occurred at precisely the same position as was found in a separate selection experiment in which compensatory mutations ameliorating harmful fitness effects associated with deletion of *rodA* were sought (E Franceschini, DW Rogers, and PB Rainey, unpublished). The prevalence of *pbp1A* mutations in the work reported here likely reflects the fact that loss-of-function mutations in *pbp1A* are more readily achieved compared to gain-of-function mutations in *ftsZ*. This

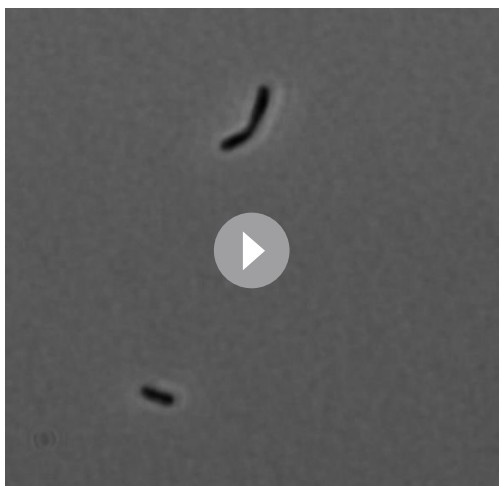

**Video 3.** Faulty septation of SBW25 Δ*mreB*
Δ*PFLU4921–4925* cells.
https://elifesciences.org/articles/98218/figures#video3

is evident in screening of mutations in *pbp1A* at generation 50, where all lines carried at least one *pbp1A* mutation (*Supplementary file 2*). It will be of interest to determine whether *ftsZ* mutations that restore fitness to Δ*rodA* cells also compensate for defects due to elimination of *mreB*.

Derived Line 7, which carries a five-gene deletion spanning the outer membrane porin OprD – and with the possibility of additional effects via mutation in *ftsE* – provides a contrasting route for compensatory evolution. Although *oprD* has not been specifically deleted, it seems reasonable to predict that in the absence of the porin, influx of water is likely limited, thus damping changes in turgor pressure and reducing cell swelling. Curiously, deletion of the focal five genes from SBW25 led to incomplete septation (*Figure 4C* and *Figure 2—figure supplement 9*), with an indication from time-resolved videos of Δ*mreB* Δ*PFLU4921–4925* (*Video 3*) that there is also fault with septation in this background. This suggests that while Δ*PFLU4921–4925* contributed to rescue of Δ*mreB*, it came at a cost to septum formation, which was subsequently rescued by mutation in the transmembrane divisome protein FtsE.

In conclusion, we return to the possibilities for new understanding of complex cellular processes arising from use of natural selection to rescue genetically perturbed cells (*Lind et al., 2019*; *Remigi et al., 2019*; *LaBar et al., 2020*). Here, a fitness-compromised, but viable, *mreB* deletion mutant of *P. fluorescens* SBW25 provided opportunity to explore consequences of both the initial genetic lesion and the bases of fitness compensation. In part attributable to the fact that Δ*mreB* SBW25 is viable, it was possible to glimpse functional effects arising from deletion of MreB, namely, uneven rates of growth among daughter cells caused by heterogeneity in patterns of cell wall synthesis. Restoration of fitness was readily achieved by mutations that reduced (or eliminated) Pbp1A function or increased septation. The primary effect was to reduce variance in growth rate among daughter cells, which is connected to resumption in homogeneity of cell wall synthesis. Disentangling two explanatory hypotheses – one invoking a role for Pbp1B and the other that elongasome function in SBW25 depends on a correct balance of among cell wall synthesising enzymes – awaits future work.

## Materials and methods
### Bacterial strains and culture conditions

*E. coli* and *P. aeruginosa* were grown at 37°C; and *P. fluorescens* SBW25 at 28°C. Antibiotics were used at the following concentrations for *E. coli* and/or *P. fluorescens* SBW25: 12 μg mL$^{-1}$ tetracycline; 30 μg mL$^{-1}$ kanamycin; 100 μg mL$^{-1}$ ampicillin. Bacteria were propagated in Lysogeny Broth (LB: 10 g tryptone, 5 g yeast extract, 10 g NaCl per litre).

### Strain construction

The Δ*mreB* and Δ*pbp1A* mutants were constructed using SOE-PCR (splicing by overlapping extension using the polymerase chain reaction), followed by a two-step allelic exchange protocol (*Rainey, 1999*). All media for construction of *mreB* mutants were supplemented with 10 mM MgCl$_2$. Genome sequencing confirmed the absence of suppressor mutations. The same procedure was used to reconstruct the mutations from the evolved lines (*pbp1A* G1450A, *pbp1A* (A1084C), Δ*pflu4921–4925* (5399934–5403214del)) and ectopic Mini-Tn*7T-mreB* into ancestral SBW25 and Δ*mreB* backgrounds. DNA fragments flanking the mutation of interest were amplified using two primer pairs. The internal primers were designed to have overlapping complementary sequences that allowed the resulting fragments to be joined together in a subsequent PCR reaction. The resulting DNA product was

TA-cloned into pCR8/GW/TOPO (Invitrogen). This was then subcloned into the pUIC3 vector, which was mobilised via conjugation into SBW25 with the help of pRK2013. Transconjugants were selected on LB plates supplemented with nitrofurantoin, tetracycline, and X-gal. Allelic exchange mutants identified as white colonies were obtained, where possible, by cycloserine enrichment to select against tetracycline-resistant cells, and tetracycline-sensitive clones were examined for the deletion or mutations using PCR and DNA sequencing. Complementation of *mreB* in the Δ*mreB* background was achieved by cloning the PCR-amplified DNA fragment into pUC18-mini-Tn7T-LAC, which was subsequently integrated into the *P. fluorescens* SBW25 genome at a unique site located downstream of the *glmS* gene (*Liu et al., 2014*).

## Growth dynamics

Strains were initially grown in 5 mL LB broth at 28°C overnight, except for Δ*mreB* which was grown for 2 days. Growth curves were started by adding 2 µL aliquots of overnight cultures to LB to a final volume of 200 µL (1:100 dilution) in 96-well microplates (three replicates). Plates were incubated at 28°C (with shaking) for 24–48 hr in a BioTek Synergy 2 plate reader. Growth was measured at an absorbance of 600 nm and recorded at 5 min intervals using the Gen5 software (Bio-Tek). Growth curves were processed as semi-log plots to identify the lag and exponential phases. Doubling times were calculated as: $t=\ln(2)/k$.

## Evolution experiment

Ten replicate populations of the Δ*mreB* strain were grown in 5 mL aliquots of LB broth at 28°C with shaking at 180 rpm. Every 24 hr, 5 µL was transferred to fresh media. Every 5 days, samples of each population were collected and stored at –80°C in 15% (vol/vol) glycerol. The number of generations per transfer changed over the course of the experiment but was approximately 10 generations per 24 hr period (100 transfers [~1000 generations] were performed).

## Competitive fitness assay

Competitive fitness was determined relative to ancestral SBW25 marked with GFP. This strain was constructed using the mini-Tn7 transposon system, expressing GFP and a gentamicin resistance marker (mini-Tn7 (Gm)P$_{rrnB}$ P1 gfp-a). Strains were grown to exponential phase in shaken LB at 28°C before beginning competitive fitness assays. Competing strains were mixed with SBW25-GFP at a 1:1 ratio by adding 150 µL of each strain to 5 mL LB and grown for 3 hr. Initial ratios were determined by counting 100,000 cells using flow cytometry (BD FACS Diva II). Suitable dilutions of the initial population were plated on LB agar plates to determine viable counts. The mixed culture was diluted 1000-fold in LB, then incubated at 28°C for 24 hr. Final viable counts and ratios were determined as described above. The number of generations over 24 hr of growth were determined using the formula ln(final population/initial population)/ln(2). Selection coefficients were calculated using the regression model $s = \frac{\ln\left(\frac{R(t)}{R(0)}\right)}{t}$, where $R$ is the ratio of the competing strain to SBW25-GFP and $t$ is time. Control experiments were conducted to determine the fitness cost of the GFP marker in SBW25. For each strain, the competition assay was performed with a minimum of three biological replicates. Ancestral SBW25 had a relative fitness of 1.0 when compared to the marked strain, indicating that the GFP insert is neutral, and that the SBW25-GFP strain was a suitable reference strain for this assay.

## Mutation detection

In order to detect mutations in the experimental lines, total DNA was extracted from the Δ*mreB* strain and from each of the 10 populations at 500 and 1000 generations. Samples were submitted for 100 bp single-end Illumina DNA sequencing at the Australian Genome Research Facility in 2012. An average of 6,802,957 single reads were obtained per sample and these were assembled against the reference *P. fluorescens* SBW25 genome (GenBank accession: OV986001.1) using the clonal option of Breseq with default parameters (*Deatherage and Barrick, 2014*). Mutations were identified with Breseq's HTML output and in the case of predicted novel junctions manually inspected with Geneious Prime (https://www.geneious.com). The raw sequence files and Breseq-generated data outputs are available at https://doi.org/10.17617/3.CU5SX1.

To detect the frequency of the *pbp1A* mutations in experimental lines at generation 50, a colony sequencing approach was used. Colony sequencing was used to prevent the confounding effects of

altered cell size, and genomic content per cell, on estimates of the frequency of mutants. To ensure sequenced colonies were isogenic, the frozen glycerol stock from generation 50 was streaked on LB agar plates and individual colonies were each re-streaked on LB agar. Resulting colonies were then used as a template for PCR, using primers that allowed amplification of all the mutations identified by Illumina Sequencing at 500 and 1000 generations. Sanger sequencing was used to identify mutations, with mutations only called if present in Sanger sequencing from both forward and reverse primers, with sequences mapped to the *pbp1A* sequence from the SBW25 reference genome using Geneious Prime.

## Microscopy

Phase contrast and fluorescence images were captured using the BX61 upright microscope (Olympus) at ×100 using a Hamamatsu Orca Flash 4.0 camera. The microscope is equipped with an HXP lamp with appropriate multiband pass filters for different channels. Cell^P/Cellsens software (Olympus) was used to control the microscope. Cells were grown in LB and harvested at log phase ($OD_{600}$ 0.4). Viability assays were conducted using the LIVE/DEAD BacLight Bacterial Viability Kit (Thermo Fisher). Viability was measured as the proportion of live cells in the total population (live/(live+dead)). Nucleoid staining was performed using the DAPI nucleic acid stain (Thermo Fisher) following the manufacturer's protocols.

## Time-lapse on agarose pads

Strains were inoculated in LB from glycerol stocks and grown with shaken overnight at 28°C. Cultures were then diluted 100-fold in fresh LB and seeded on a gel pad (1% agarose in LB). The preparation was sealed on a glass coverslip with double-sided tape (Gene Frame, Fischer Scientific). A duct was cut through the center of the pad to allow for oxygen diffusion into the gel. Temperature was maintained at 30°C using a custom-made temperature controller. Bacteria were imaged on a custom-built microscope using a ×100/NA 1.4 objective lens (Apo-ph3, Olympus) and an Orca-Flash4.0 CMOS camera (Hamamatsu). Image acquisition and microscope control were actuated with a LabView interface (National Instruments) (*Julou et al., 2013*). Typically, we monitored 10 different locations; images were taken every 5 min in correlation mode. Segmentation and cell lineage were computed using a MATLAB code implemented from Schnitzcell (*Locke and Elowitz, 2009*). Bacteria were tracked for three generations. In order to limit the number of generations, we monitored bacteria harvested from a liquid stationary phase. Since bacteria shorten after entering stationary phase, bacteria lengthened when resuming growth under the microscope, so that the elongation rate is larger than the septation rate for the first generations (*Figure 3B*). To obtain the control line in our experimental configuration, we measured these rates in three species (ancestral *P. fluorescens* SBW25, *E. coli* MG1655, and *P. aeruginosa* PAO1). Points below the line indicate a reduction of cell volume beyond the fact that cells simply resumed growth.

## Scanning electron microscopy

Cells were grown in LB and harvested in log phase. Cells were fixed in modified Karnovsky's fixative then placed between two membrane filters (0.4 µm, Isopore, Merck Millipore LTD) in an aluminum clamp. Following three washes of phosphate buffer, the cells were dehydrated in a graded-ethanol series, placed in liquid $CO_2$, then dried in a critical-point drying chamber. The samples were mounted onto aluminum stubs and sputter coated with gold (BAL-TEC SCD 005 sputter coater) and viewed in a FEI Quanta 200 scanning electron microscope at an accelerating voltage of 20 kV.

## Image analysis

Compactness and estimated volume measurements of cells from liquid culture: The main measure of cell shape, compactness or $C$, was computed by the CMEIAS (*Liu et al., 2001*) software as: $\frac{\sqrt{4Area/\pi}}{major\ axis\ lenght}$ (*Liu et al., 2001*). Estimated volume or $V_e$ was estimated with different formula, according to cell compactness, for spherical cells that have a compactness ≥0.7, $V_e$ was computed using the general formula for ellipsoids: $V = \frac{4}{3}(L/2)(W/2)^2$, where $L$ = length and $W$ = width. $V_e$ of rod-shaped cells, defined as having a compactness value ≤0.7, were computed using the combined formulas for cylinders and spheres: $V_e = (W/2)^2(L - W) + \frac{4}{3}(W/2)^3$.

Cell volume and division axis of cells on agarose pads. Cell volume was computed from the mask retrieved after image segmentation. Masks were fitted to an ellipse. The elongation axis is given by the direction of the major axis. The division axis relative to the mother was computed by comparing the elongation axis between mother and sister cells using the following formula: $|\sin\theta|$, where $\theta$ is the angle between mother and sister cells. $|\sin\theta| = 0$ means that mother and daughter cells are aligned, $|\sin\theta| = 1$ means that they are perpendicular.

## Probability p to pass to the next generation

For the second generation on the agar pad, we computed the probability to pass to the next generation as the capability of cells to progress through the cell cycle and divide. Bacteria that did not grow or stopped elongating before dividing were classified as non-proliferating. For all non-proliferating bacteria, we confirmed that no division occurred during the next 5 hr. The $p$ ratio was then simply computed as the number of viable daughter cells divided by the expected number. Error on the probability is estimated at the 95% confidence interval by the formula: $1.96\sqrt{(p(1-p)/N)}$ , where $N$ is the total number of cells.

## Elongation rate, septation rate, growth asymmetry, and the precision of septum positioning

The elongation rate, $r$, measured in min$^{-1}$ is the rate at which cell volume increases: $V(t) = V_0 e^{rt}$, where $t$ is the relative time in the cell cycle. The septation rate is given by $\frac{\ln 2}{\tau}$ , where $\tau$ is the average division time, i.e., the time between birth and septation. For all sister cell pairs, we computed relative difference in growth between sister cells by measuring the relative elongation rate (i.e. volume extension rate) of sister cells at generation F1: $|(r_1 - r_2)/(r_1 + r_2)|$, where $r_{1,2}$ is the elongation rate for each cell of sister pairs. The population average $C_g = \langle |r1 - r2|/(r1 + r2)\rangle_{cells}$ was computed on the proliferating sub-population in order to avoid trivial bias due to cell proliferation arrest of one of the two sister cells. We did the same for the relative difference in division time and measured $C_T = \langle |T1 - T2|/(T1 + T2)\rangle_{cells}$, where $T_{1,2}$ are the division time of sister cells. For all cells at generation F1, we computed the precision of septum positioning by measuring the relative cell volume of the two daughter cells: $|(V_1 - V_2)/(V_1 + V_2)|$, where $V_{1,2}$ are the volumes of each daughter cell. The population average $C_s = \langle |V1 - V2|/(V1 + V2)\rangle_{cells}$ was computed for every F1 cell having offspring.

### Laplace argument

For a given geometry, cell volume is determined, at steady state, by an analogue of the Young-Laplace law for thin elastic membranes, which relates the pressure discontinuity $\Delta P$ at the cell wall to membrane tension $T$ and radius of curvature $R$. The pressure jump is powered by turgor pressure in the cytoplasm. For spheres, having two identical curvature radii $R_s$, the law is : $\Delta P_s = 2\frac{T_s}{R_s}$ . For cylinders, having only one finite radius of curvature $R_c$, the law is $\Delta P_c = \frac{T_c}{R_c}$ . Assuming that turgor pressure and membrane tension are the same, $\Delta P_s = \Delta P_c$ and $T_s = T_c$ , we obtain $R_s = 2R_c$. Hence, the ratio of volumes between the two geometries can be expressed as a function of the original ancestral SBW25 length $L$ and width $2R_c$: $\frac{V_s}{V_c} = \frac{2^5 R_c}{3L}$ , where $V_s$ and $V_c$ are the volume of the sphere and the cylinder, respectively. Hence, a change from cylindrical to spherical shape should result in a volume increase by a factor 1.78 for *P. fluorescens*, with a diameter of 1 µm and a length of 3 µm (*Figure 3A*).

### Protein sequence alignment and modelling

Protein sequences were obtained from NCBI BLAST (http://blast.ncbi.nlm.nih.gov) and The Pseudomonas Genome Database (*Winsor et al., 2016*), and aligned using MEGA7 (*Kumar et al., 2016*). The sequence alignment was visualised using ESPript 'Easy Sequencing in PostScript' (*Robert and Gouet, 2014*). Protein visualisation was performed using visual molecular dynamics (*Humphrey et al., 1996*) based on the crystal structure of *A. baumannii* Pbp1a in complex with Aztreonam as the base model (Pbp1A from *A. baumannii* shares a 73% amino acid sequence identity [E value = 0.0] to the Pbp1A of *P. fluorescens* SBW25). Sequences were aligned, and locations of the mutations in the evolved lines were mapped in the corresponding regions. The PDB file was downloaded from the RCSB Protein Data Bank (http://www.rcsb.org/) using PDB ID 3UE0.

## Membrane preparation and detection of PBPs

For the membrane preparation of the different genotypes of *P. fluorescens* SBW25, 150 mL cell cultures were grown to an $OD_{600}$ of 0.6 and harvested by centrifugation at $15,000 \times g$ for 15 min at 4°C (Beckman JLA-16.250 rotor, Beckman Avanti J-25, Beckman Coulter). Pellets were washed with 20 mM potassium phosphate (pH 7.5) and 140 mM sodium chloride buffer. Following another 30 min of centrifugation at $360 \times g$ at 4°C, cells were disrupted by passing twice through a French press ('Pressure Cell' Homogenizer FPG12800, Stansted Fluid Power Ltd). Cell lysates were subjected to centrifugation at $360 \times g$ for 10 min at 4°C. Supernatant fractions were collected and centrifuged at $75,600 \times g$ for 90 min at 4°C (Beckman JA25.50 rotor, Beckman Avanti J-25, Beckman Coulter). The resulting pellets were washed once again and resuspended in 100 µL potassium phosphate buffer. The concentration of the membrane preparations was measured using Bradford reagent (Bio-Rad) with bovine serum albumin as a standard (Sigma).

For detection of PBPs, membranes were incubated for 30 min at 37°C with a fluorescent labelling agent, Bocillin FL Penicillin (Invitrogen, Thermo Fisher Scientific) (final concentration: 0.05 mM). Finally, the samples were denatured with 5× SDS loading buffer with β-mercaptoethanol as a reducing agent at 100°C for 5 min, followed by centrifugation to remove membrane debris. 20 µL of reaction mixture was subjected to SDS-PAGE analysis on an 8% polyacrylamide gel, protected from the light. Bocillin FL-labelled PBPs were visualised with a laser scanner, Typhoon FLA 9500 (GE Healthcare Life Sciences) at 473 nm with a 530DF20 emission filter.

## Peptidoglycan isolation and analysis

For peptidoglycan profiling and analysis, *P. fluorescens* genotypes were grown in LB medium to an $OD_{600}$ of 0.2 and harvested. Purification of peptidoglycan was carried out as previously described (*Desmarais et al., 2013*) with some minor changes. Briefly, the cell pellets were boiled in an equal volume of 5% (wt/vol) SDS for approximately an hour and left stirring overnight at room temperature. The sacculi were washed repeatedly with MilliQ water by ultracentrifugation ($150,000 \times g$, 15 min, 20°C TLA100.3 Beckman rotor; Optima Max ultracentrifuge Beckman, Beckman Coulter) until SDS was eliminated from the samples. The clean sacculi were digested with muramidase (Cellosyl 100 µg mL$^{-1}$) overnight at 37°C. The enzyme reaction was stopped by heat inactivation at 100°C. Coagulated proteins were removed by centrifugation ($21,000 \times g$ for 15 min, Heraeus Pico 21 Microcentrifuge, Thermo Fisher Scientific). For sample reduction, supernatants were adjusted to pH 8.5–9.0 with borate buffer, followed by incubation for 30 min at room temperature with freshly prepared NaBH$_4$ solution (final concentration of 10 mg mL$^{-1}$). Finally, the pH was adjusted to 3.5 by the addition of phosphoric acid. Muropeptides were separated by UPLC on a Waters UPLC system (Waters) equipped with a Kinetex C18 UPLC Column, 130 Å, 1.7 µm, 2.1 mm × 150 mm (Phenomenex) and a dual wavelength absorbance detector. Elution of muropeptides was detected at 204 nm. Separation of the muropeptides was carried out at 45°C using a linear gradient from Buffer A (formic acid 0.1% [vol/vol]) to Buffer B (formic acid 0.1% [vol/vol], acetonitrile 40% [vol/vol]) in an 18 min run with 0.250 mL min$^{-1}$ flow.

## Fluorescent D-amino acid labelling

Short-pulse staining of FDAA incorporation into growing *P. fluorescens* cell walls was accomplished using FDAA BODIPY-FL 3-amino-D-alanine (BADA) (*Hsu et al., 2017*) (green; emission 502 nm, excitation 518 nm). Briefly, BADA was diluted in dimethyl sulfoxide to a concentration of 100 mM. For each test strain, 1 mL of exponential culture ($OD_{600}$ of 0.4) was pelleted by centrifugation ($10,000 \times g$:$2000 \times g$ for Δ*mreB*) and resuspended in 100 µL. BADA was added to achieve a final concentration of 1 mM BADA. Incubation at 28°C for 20% of strain generation time. Following this, 230 µL of ice-cold 100% ethanol was added to culture. An additional 1 mL of ice-cold 70% ethanol was added to remove excess dye. Cells were fixed on ice for 15 min before being washed with 1 mL of PBS three times to remove excess BADA. Finally, cells were resuspended in 20–100 µL of PBS and subjected to standard fluorescent microscopy on agarose pads. Fluorescent microscopy was performed as described above.

## Acknowledgements

We thank Olin Silander for helpful discussions and for his assistance to PRY with principal components analysis of cell shape, Sebastian Schmeier and Saumya Agrawal for bioinformatic analyses, Dave Rogers for valuable discussion, and both Dave Rogers and Tim Cooper for comments on the manuscript. PBR is especially grateful to Malavika Venu, Norma Rivera, and Dave Rogers who made the Δ*mreB* Δ*pbp1A* mutant and Ellen McConnell who introduced fluorescent markers and measured fitness effects; Carsten Fortmann-Grote for assistance with collation and posting of primary data to Edmond. Electron Microscopy was provided by Massey University and was performed by Niki Murray, Manawatu Microscopy and Imaging Centre, Massey University, Palmerston North, NZ. PBR thanks the Royal Society of New Zealand (James Cook Fellowship) and acknowledges generous core support from the Max Planck Society. ND thanks the IdEx Université de Paris (ANR-18-IDEX-0001), the "Institut Pierre-Gilles de Gennes" ("Investissements d'Avenir" program ANR-10-IDEX-0001-02 PSL and ANR-10-LABX-31) and the Qlife Institute of Convergence (PSL). Research in the Cava lab was supported by the Swedish Research Council (2018-02823 and 2018-05882), Umeå University, the Knut and Alice Wallenberg Foundation and the Kempe Foundation (SMK2062). Open access funding is provided by the Max Planck Society. PBR dedicates the paper to the memory of Andrew Spiers, colleague and ardent wrinkliologist, whose 2002 manuscript first reported the fact that a transposon insertion in mreB abolishes wrinkly morphology, but not viability.

## Additional information

### Competing interests

Paul B Rainey: Reviewing editor, eLife. The other authors declare that no competing interests exist.

### Funding

| Funder | Grant reference number | Author |
|---|---|---|
| Royal Society of New Zealand | James Cook Fellowship | Paul B Rainey |
| Max Planck Society | Core support | Paul B Rainey |
| Swedish Research Council | 2018-02823 | Felipe Cava |
| Swedish Research Council | 2018-05882 | Felipe Cava |
| Umeå University | | Felipe Cava |
| Knut and Alice Wallenberg Foundation | | Felipe Cava |
| Kempe Foundation | SMK2062 | Felipe Cava |
| Agence Nationale de la Recherche | ANR-19-CE13-0026 - ADOBE | Nicolas Desprat |

The funders had no role in study design, data collection and interpretation, or the decision to submit the work for publication. Open access funding provided by Max Planck Society.

### Author contributions

Paul Richard J Yulo, Investigation, Construction of SBW25 GFP, phase contrast and fluorescence (BADA) microscopy, flow cytometry and growth assays; Nicolas Desprat, Data curation, Formal analysis, Investigation, Visualization, Methodology, Writing – original draft, Writing – review and editing, Microscopy, data collection/analysis; Monica L Gerth, Investigation, Writing – review and editing, Construction of SBW25 ΔmreB, selection experiment and fitness measures; Barbara Ritzl-Rinkenberger, Formal analysis, Investigation, Visualization, Writing – review and editing, Peptidoglycan analysis, Bocillin-FL PBP labelling assays and data analysis; Andrew D Farr, Investigation, Writing – review and editing, Sequencing and data analysis; Yunhao Liu, Investigation, Genetics and mutation construction; Xue-Xian Zhang, Formal analysis, Supervision, Investigation, Methodology, Writing – review and editing,

Genetics and mutation construction; Michael Miller, Investigation, Mutant construction, aztreonam analysis; Felipe Cava, Supervision, Methodology, Writing – review and editing, Supervision and data analysis; Paul B Rainey, Conceptualization, Resources, Data curation, Formal analysis, Supervision, Funding acquisition, Validation, Investigation, Writing – original draft, Project administration, Writing – review and editing; Heather L Hendrickson, Supervision, Writing – original draft, Writing – review and editing, DNA extraction and analysis

#### Author ORCIDs
Paul Richard J Yulo ⓘD https://orcid.org/0009-0008-2288-0711
Nicolas Desprat ⓘD https://orcid.org/0000-0002-5016-9360
Felipe Cava ⓘD https://orcid.org/0000-0001-5995-718X
Paul B Rainey ⓘD https://orcid.org/0000-0003-0879-5795
Heather L Hendrickson ⓘD https://orcid.org/0000-0003-3471-4397

Reviewer #1 (Public review): https://doi.org/10.7554/eLife.98218.4.sa1
Reviewer #3 (Public review): https://doi.org/10.7554/eLife.98218.4.sa2
Author response https://doi.org/10.7554/eLife.98218.4.sa3

## Additional files

### Supplementary files
Supplementary file 1. Mutations identified in derived lines at generations 500 and 1000.

Supplementary file 2. Mutations in *pbp1A* at generation 50.

MDAR checklist

### Data availability
All primary data are available on Edmond at this DOI https://doi.org/10.17617/3.CU5SX1.

The following dataset was generated:

| Author(s) | Year | Dataset title | Dataset URL | Database and Identifier |
|---|---|---|---|---|
| Yulo et al | 2025 | Evolutionary rescue of spherical mreB deletion mutants of the rod-shape bacterium Pseudomonas fluorescens SBW25 | https://doi.org/10.17617/3.CU5SX1 | Edmond, 10.17617/3.CU5SX1 |

The following previously published dataset was used:

| Author(s) | Year | Dataset title | Dataset URL | Database and Identifier |
|---|---|---|---|---|
| Han S | 2011 | Crystal structure of Acinetobacter baumannii PBP1a in complex with Aztreonam | https://www.rcsb.org/structure/3UE0 | RCSB Protein Data Bank, 3UE0 |

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
