## [Editor Report · eLife Assessment]

This **important** study combines **convincing** evolution experiments with molecular and genetic techniques to study how a genetic lesion in MreB that causes rod-shaped cells to become spherical, with concomitant deleterious fitness effects, can be rescued by natural selection. The detailed mechanistic investigation increases our understanding of how mreB contributes to cell wall synthesis and shows how compensatory mutations may reestablish its homogeneity.

---

## [Referee Report · Reviewer #1 (Public review)]

Summary:

The authors performed experimental evolution of MreB mutants that have a slow growing round phenotype and studied the subsequent evolutionary trajectory using analysis tool from molecular biology. It was remarkable and interesting that they found that the original phenotype was not restored (most common in these studies) but that the round phenotype was maintained.

Strengths:

The finding that the round phenotype was maintained during evolution rather than that the original phenotype, rod shape cells, was recovered is interesting. The paper extensively investigates what happens during adaptation with various different techniques. Also the extensive discussion of the findings at the end of the paper is well thought through and insightful.

---

## [Referee Report · Reviewer #3 (Public review)]

This paper addresses a long-standing problem in microbiology: the evolution of bacterial cell shape. Bacterial cells can take a range of forms, among the most common being rods and spheres. The consensus view is that rods are the ancestral form and spheres the derived form. The molecular machinery governing these different shapes is fairly well understood but the evolutionary drivers responsible for the transition between rods and spheres is not. Enter Yulo et al.'s work. The authors start by noting that deletion of a highly conserved gene called MreB in the Gram-negative bacterium Pseudomonas fluorescens reduces fitness but does not kill the cell (as happens in other species like E. coli and *B. subtilis*) and causes cells to become spherical rather than their normal rod shape. They then ask whether evolution for 1000 generations restores the rod shape of these cells when propagated in a rich, benign medium.

The answer is no. The evolved lineages recovered fitness by the end of the experiment, growing just as well as the unevolved rod-shaped ancestor, but remained spherical. The authors provide an impressively detailed investigation of the genetic and molecular changes that evolved. Their leading results are:

(1) The loss of fitness associated with MreB deletion causes high variation in cell volume among sibling cells after cell division;

(2) Fitness recovery is largely driven by a single, loss-of-function point mutation that evolves within the first ~250 generations that reduces the variability in cell volume among siblings;

(3) The main route to restoring fitness and reducing variability involves loss of function mutations causing a reduction of TPase and peptidoglycan cross-linking, leading to a disorganized cell wall architecture characteristic of spherical cells.

The inferences made in this paper are on the whole well supported by the data. The authors provide a uniquely comprehensive account of how a key genetic change leads to gains in fitness and the spectrum of phenotypes that are impacted and provide insight into the molecular mechanisms underlying models of cell shape.

---

## [Author Response]

[The following is the authors’ response to the previous reviews.]

As to the exceptionally minor issue, namely, correction for multiple statistical tests (minor because the data and the error are presented in the text). We have now conducted one-way ANOVA to back the data displayed in Fig 4A., and Supp. Figs 19 and 21. In each case ANOVA revealed a highly significant difference among means: Dunnett’s post hoc test was then used to test each result against SBW25, with the multiple comparisons corrected for in the analysis.

This resulted in changes to the description of the statistical analysis in the following captions:

To Figure 4.

Where we previously referred to paired t-tests we now state: ANOVA revealed a highly significant difference among means [*F*_7,16_ = 8.19, *p* < 0.001] with Dunnett’s post-hoc test adjusted for multiple comparisons showing that five genotypes (*) differ significantly (*p* < 0.05) from SBW25.

To Supplementary Figure 19.

Where we previously referred to paired t-tests we now state: ANOVA revealed a highly significant difference among means [*F*_7,16_ = 16.74, *p* < 0.001] with Dunnett’s post-hoc test adjusted for multiple comparisons showing that three genotypes (*) differ significantly (*p* < 0.05) from SBW25.

To Supplementary Figure 21.

Where we previously referred to paired t-tests we now state: ANOVA revealed a highly significant difference among means [*F*_7,89_ = 9.97, *p* < 0.0001] with Dunnett’s post-hoc test adjusted for multiple comparisons showing that SBW25 *∆mreB* and SBW25 ∆PFLU4921-4925 are significantly different (*) from SBW25 (*p* < 0.05).

[The following is the authors’ response to the original reviews.]

**Public Reviews:**

**Reviewer #1 (Public Review):**
Summary:The authors performed experimental evolution of MreB mutants that have a slow-growing round phenotype and studied the subsequent evolutionary trajectory using analysis tools from molecular biology. It was remarkable and interesting that they found that the original phenotype was not restored (most common in these studies) but that the round phenotype was maintained.Strengths:The finding that the round phenotype was maintained during evolution rather than that the original phenotype, rod-shaped cells, was recovered is interesting. The paper extensively investigates what happens during adaptation with various different techniques. Also, the extensive discussion of the findings at the end of the paper is well thought through and insightful.Weaknesses:I find there are three general weaknesses:(1) Although the paper states in the abstract that it emphasizes "new knowledge to be gained" it remains unclear what this concretely is. On page 4 they state 3 three research questions, these could be more extensively discussed in the abstract. Also, these questions read more like genetics questions while the paper is a lot about cell biological findings.

Thank you for drawing attention to the unnecessary and gratuitous nature of the last sentence of the Abstract. We are in agreement. It has been modified, and we have taken advantage of additional word space to draw attention to the importance of the two competing (testable) hypotheses laid out in the Discussion.

As to new knowledge, please see the Results and particularly the Discussion. But beyond this, and as recognised by others, there is real value for cell biology in seeing how (and whether) selection can compensate for effects that are deleterious to fitness. The results will very often depart from those delivered from, for example, suppressor analyses, or bottom up engineering.

In the work recounted in our paper, we chose to focus – by way of proof-of principle – on the most commonly observed mutations, namely, those within *pbp1A*. But beyond this gene, we detected mutations in other components of the cell shape / division machinery whose connections are not yet understood and which are the focus of on-going investigation.

As to the three questions posed at the end of the Introduction, the first concerns whether selection can compensate for deleterious effects of deleting *mreB* (a question that pertains to evolutionary aspects); the second seeks understanding of genetic factors; the third aims to shed light on the genotype-to-phenotype map (which is where the cell biology comes into play). Given space restrictions, we cannot see how we could usefully expand, let alone discuss, the three questions raised at the end of the Introduction in restrictive space available in the Abstract.

(2) It is not clear to me from the text what we already know about the restoration of MreB loss from suppressors studies (in the literature). Are there suppressor screens in the literature and which part of the findings is consistent with suppressor screens and which parts are new knowledge?

As stated in the Introduction, a previous study with *B. subtilis* (which harbours three MreB isoforms and where the isoform named “MreB” is essential for growth under normal conditions), suppressors of MreB lethality were found to occur in *ponA*, a class A penicillin binding protein (Kawai et al., 2009). This led to recognition that MreB plays a role in recruiting Pbp1A to the lateral cell wall. On the other hand, Patel et al. (2020) have shown that deletion of classA PBPs leads to an up-regulation of rod complex activity. Although there is a connection between rod complex and class A PBPs, a further study has shown that the two systems work semi-autonomously (Cho et al., 2016).

Our work confirms a connection between MreB and Pbp1A, and has shed new light on how this interaction is established by means of natural selection, which targets the integrity of cell wall. Indeed, the Rod complex and class A PBPs have complementary activities in the building of the cell wall with each of the two systems able to compensate for the other in order to maintain cell wall integrity. Please see the major part of the Discussion. In terms of specifics, the connection between *mreB* and *pbp1A* (shown by Kawai et al (2009)) is indirect because it is based on extragenic transposon insertions. In our study, the genetic connection is mechanistically demonstrated. In addition, we capture that the evolutionary dynamics is rapid and we finally enriched understanding of the genotype-to-phenotype map.

(3) The clarity of the figures, captions, and data quantification need to be improved.

Modifications have been implemented. Please see responses to specific queries listed below.

**Reviewer #2 (Public Review):**
Yulo et al. show that deletion of MreB causes reduced fitness in P. fluorescens SBW25 and that this reduction in fitness may be primarily caused by alterations in cell volume. To understand the effect of cell volume on proliferation, they performed an evolution experiment through which they predominantly obtained mutations in pbp1A that decreased cell volume and increased viability. Furthermore, they provide evidence to propose that the pbp1A mutants may have decreased PG cross-linking which might have helped in restoring the fitness by rectifying the disorganised PG synthesis caused by the absence of MreB. Overall this is an interesting study.Queries:Do the small cells of mreB null background indeed have no DNA? It is not apparent from the DAPI images presented in Supplementary Figure 17. A more detailed analysis will help to support this claim.

It is entirely possible that small cells have no DNA, because if cell division is aberrant then division can occur prior to DNA segregation resulting in cells with no DNA. It is clear from microscopic observation that both small and large cells do not divide. It is, however, true, that we are unable to state – given our measures of DNA content – that small cells have no DNA. We have made this clear on page 13, paragraph 2.

What happens to viability and cell morphology when pbp1A is removed in the mreB null background? If it is actually a decrease in pbp1A activity that leads to the rescue, then pbp1A- mreB- cells should have better viability, reduced cell volume and organised PG synthesis. Especially as the PG cross-linking is almost at the same level as the T362 or D484 mutant.

Please see fitness data in Supp. Fig. 13. Fitness of ∆*mreB* ∆*pbp1A* is no different to that caused by a point mutation. Cells remain round.

What is the status of PG cross-linking in ΔmreB Δpflu4921-4925 (Line 7)?

This was not analysed as the focus of this experiment was PBPs. *A priori*, there is no obvious reason to suspect that ∆4921-25 (which lacks *oprD*) would be affected in PBP activity.

What is the morphology of the cells in Line 2 and Line 5? It may be interesting to see if PG cross-linking and cell wall synthesis is also altered in the cells from these lines.

The focus of investigation was restricted to L1, L4 and L7. Indeed, it would be interesting to look at the mutants harbouring mutations in *:sZ*, but this is beyond scope of the present investigation (but is on-going). The morphology of L2 and L5 are shown in Supp. Fig. 9.

The data presented in 4B should be quantified with appropriate input controls.

Band intensity has now been quantified (see new Supp. Fig .20). The controls are SBW25, SBW25∆*pbp1A*, SBW25 ∆*mreB* and SBW25 ∆*mreBpbp1A* as explained in the paper.

What are the statistical analyses used in 4A and what is the significance value?

Our oversight. These were reported in Supp. Fig. 19, but should also have been presented in Fig. 4A. Data are means of three biological replicates. The statistical tests are comparisons between each mutant and SBW25, and assessed by paired *t*-tests.

A more rigorous statistical analysis indicating the number of replicates should be done throughout.

We have checked and made additions where necessary and where previously lacking. In particular, details are provided in Fig. 1E, Fig. 4A and Fig. 4B. For Fig. 4C we have produced quantitative measures of heterogeneity in new cell wall insertion. These are reported in Supp. Fig. 21 (and referred to in the text and figure caption) and show that patterns of cell wall insertion in ∆*mreB* are highly heterogeneous.

**Reviewer #3 (Public Review):**
This paper addresses an understudied problem in microbiology: the evolution of bacterial cell shape. Bacterial cells can take a range of forms, among the most common being rods and spheres. The consensus view is that rods are the ancestral form and spheres the derived form. The molecular machinery governing these different shapes is fairly well understood but the evolutionary drivers responsible for the transition between rods and spheres are not. Enter Yulo et al.'s work. The authors start by noting that deletion of a highly conserved gene called MreB in the Gram-negative bacterium Pseudomonas fluorescens reduces fitness but does not kill the cell (as happens in other species like *E. coli* and *B. subtilis*) and causes cells to become spherical rather than their normal rod shape. They then ask whether evolution for 1000 generations restores the rod shape of these cells when propagated in a rich, benign medium.The answer is no. The evolved lineages recovered fitness by the end of the experiment, growing just as well as the unevolved rod-shaped ancestor, but remained spherical. The authors provide an impressively detailed investigation of the genetic and molecular changes that evolved. Their leading results are:(1) The loss of fitness associated with MreB deletion causes high variation in cell volume among sibling cells after cell division.(2) Fitness recovery is largely driven by a single, loss-of-function point mutation that evolves within the first ~250 generations that reduces the variability in cell volume among siblings.(3) The main route to restoring fitness and reducing variability involves loss of function mutations causing a reduction of TPase and peptidoglycan cross-linking, leading to a disorganized cell wall architecture characteristic of spherical cells.The inferences made in this paper are on the whole well supported by the data. The authors provide a uniquely comprehensive account of how a key genetic change leads to gains in fitness and the spectrum of phenotypes that are impacted and provide insight into the molecular mechanisms underlying models of cell shape.Suggested improvements and clarifications include:(1) A schematic of the molecular interactions governing cell wall formation could be useful in the introduction to help orient readers less familiar with the current state of knowledge and key molecular players.

We understand that this would be desirable, but there are numerous recent reviews with detailed schematics that we think the interested reader would be better consulting. These are referenced in the text.

(2) More detail on the bioinformatics approaches to assembling genomes and identifying the key compensatory mutations are needed, particularly in the methods section. This whole subject remains something of an art, with many different tools used. Specifying these tools, and the parameter settings used, will improve transparency and reproducibility, should it be needed.

We overlooked providing this detail, which has now been corrected by provision of more information in the Materials and Methods. In short we used Breseq, the clonal option, with default parameters. Additional analyses were conducted using Genieous. The BreSeq output files are provided https://doi.org/10.17617/3.CU5SX1 (which include all read data).

(3) Corrections for multiple comparisons should be used and reported whenever more than one construct or strain is compared to the common ancestor, as in Supplementary Figure 19A (relative PG density of different constructs versus the SBW25 ancestor).

The data presented in Supp Fig 19A (and Fig 4A) do not involve multiple comparisons. In each instance the comparison is between SBW25 and each of the different mutants. A paired *t*-test is thus appropriate.

(4) The authors refrain from making strong claims about the nature of selection on cell shape, perhaps because their main interest is the molecular mechanisms responsible. However, I think more can be said on the evolutionary side, along two lines. First, they have good evidence that cell volume is a trait under strong stabilizing selection, with cells of intermediate volume having the highest fitness. This is notable because there are rather few examples of stabilizing selection where the underlying mechanisms responsible are so well characterized. Second, this paper succeeds in providing an explanation for how spherical cells can readily evolve from a rod-shaped ancestor but leaves open how rods evolved in the first place. Can the authors speculate as to how the complex, coordinated system leading to rods first evolved? Or why not all cells have lost rod shape and become spherical, if it is so easy to achieve? These are important evolutionary questions that remain unaddressed. The manuscript could be improved by at least flagging these as unanswered questions deserving of further attention.

These are interesting points, but our capacity to comment is entirely speculative. Nonetheless, we have added an additional paragraph to the Discussion that expresses an opinion that has yet to receive attention:

“Given the complexity of the cell wall synthesis machinery that defines rod-shape in bacteria, it is hard to imagine how rods could have evolved prior to cocci. However, the cylindrical shape offers a number of advantages. For a given biomass (or cell volume), shape determines surface area of the cell envelope, which is the smallest surface area associated with the spherical shape. As shape sets the surface/volume ratio, it also determines the ratio between supply (proportional to the surface) and demand (proportional to cell volume). From this point of view, it is more efficient to be cylindrical (Young 2006). This also holds for surface attachment and biofilm formation (Young 2006). But above all, for growing cells, the ratio between supply and demand is constant in rod shaped bacteria, whereas it decreases for cocci. This requires that spherical cells evolve complex regulatory networks capable of maintaining the correct concentration of cellular proteins despite changes in surface/volume ratio. From this point of view, rod-shaped bacteria offer opportunities to develop unsophisticated regulatory networks.”

why not all cells have lost rod shape and become spherical.

Please see Kevin Young’s 2006 review on the adaptive significance of cell shape.

The value of this paper stems both from the insight it provides on the underlying molecular model for cell shape and from what it reveals about some key features of the evolutionary process. The paper, as it currently stands, provides more on which to chew for the molecular side than the evolutionary side. It provides valuable insights into the molecular architecture of how cells grow and what governs their shape. The evolutionary phenomena emphasized by the authors - the importance of loss-of-function mutations in driving rapid compensatory fitness gains and that multiple genetic and molecular routes to high fitness are often available, even in the relatively short time frame of a few hundred generations - are well-understood phenomena and so arguably of less broad interest. The more compelling evolutionary questions concern the nature and cause of stabilizing selection (in this case cell volume) and the evolution of complexity. The paper misses an opportunity to highlight the former and, while claiming to shed light on the latter, provides rather little useful insight.

Thank you for these thoughts and comments. However, we disagree that the experimental results are an overlooked opportunity to discuss stabilising selection. Stabilising selection occurs when selection favours a particular phenotype causing a reduction in underpinning population-level genetic diversity. This is not happening when selection acts on SBW25 ∆*mreB* leading to a restoration of fitness. Driving the response are biophysical factors, primarily the critical need to balance elongation rate with rate of septation. This occurs without any change in underlying genetic diversity.

**Recommendations for the authors:**

**Reviewer 1 (Recommendations for the Authors):**
Hereby my suggestion for improvement of the quantification of the data, the figures, and the text.- p 14, what is the unit of elongation rate?

At first mention we have made clear that the unit is given in minutes^-1

- p 14, please give an error bar for both p=0.85 and f=0.77, to be able to conclude they are different

Error on the probability p is estimated at the 95% confidence interval by the formula: 1.96(p(1−p)/N),where N is the total number of cells. This has been added in the paragraph p »probability » of the Image Analysis section in the Material and Methods.

We also added errors on p measurement in the main text.

- p 14, all the % differences need an errorbar

The error bars and means are given in Fig 3C and 3D.

- Figure 1B adds units to compactness, and what does it represent? Is the cell size the estimated volume (that is mentioned in the caption)? Shouldn't the datapoints have error bars?

Compactness is defined in the “Image Analysis” section of the Material and Methods. It is a dimensionless parameter. The distribution of individual cell shapes / sizes are depicted in Fig 1B. Error does arise from segmentation, but the degree of variance (few pixels) is much smaller than the representations of individual cells shown.

- Figure 1C caption, are the 50.000 cells?

Correct. Figure caption has been altered.

- Figure 1D, first the elongation rate is described as a volume per minute, but now, looking at the units it is a rate, how is it normalized?

Elongation rate is explained in the Materials and Methods (see the image analysis section) and is not volume per minute. It is dV/dt = r*V (the unit of r is min^-1). Page 9 includes specific mention of the unit of *r*.

- Figure 1E, how many cells (n) per replicate?

Our apologies. We have corrected the figure caption that now reads:

“Proportion of live cells in ancestral SBW25 (black bar) and *ΔmreB* (grey bar) based on LIVE/DEAD BacLight Bacterial Viability Kit protocol. Cells were pelleted at 2,000 x g for 2 minutes to preserve *ΔmreB* cell integrity. Error bars are means and standard deviation of three biological replicates (n>100).”

- Figure 1G, how does this compare to the wildtype

The volume for wild type SBW25 is 3.27µm^3 (within the “white zone”). This is mentioned in the text.

- Figure 2B, is this really volume, not size? And can you add microscopy images?

The x-axis is volume (see Materials and Methods, subsection image analysis). Images are available in Supp. Fig. 9.

- Figure 3A what does L1, L4 and L7 refer too? Is it correct that these same lines are picked for WT and delta_mreB

Thank you for pointing this out. This was an earlier nomenclature. It was shorthand for the mutants that are specified everywhere else by genotype and has now been corrected.

- Figure 3c: either way write out p, so which probability, or you need a simple cartoon that is plotted.

The value *p* is the probability to proceed to the next generation and is explained in Materials and Methods subsection image analysis. We feel this is intuitive and does not require a cartoon. We nonetheless added a sentence to the Materials and Methods to aid clarity.

- Figure 4B can you add a ladder to the gel?

No ladder was included, but the controls provide all the necessary information. The band corresponding to PBP1A is defined by presence in SBW25, but absence in SBW25 ∆*pbp1A*.

- Figure 4c, can you improve the quantification of these images? How were these selected and how well do they represent the community?

We apologise for the lack of quantitative description for data presented in Fig 4C. This has now been corrected. In brief, we measured the intensity of fluorescent signal from between 10 and 14 cells and computed the mean and standard deviation of pixel intensity for each cell. To rule out possible artifacts associated with variation of the mean intensity, we calculated the ratio of the standard deviation divided by the square root of the mean. These data reveal heterogeneity in cell wall synthesis and provide strong statistical support for the claim that cell wall synthesis in ∆*mreB* is significantly more heterogeneous than the control. The data are provided in new Supp. Fig. 21.

Minor comments:- It would be interesting if the findings of this experimental evolution study could be related to comparative studies (if these have ever been executed).

Little is possible, but Hendrickson and Yulo published a portion of the originally posted preprint separately. We include a citation to that paper.

- p 13, halfway through the page, the second paragraph lacks a conclusion, why do we care about DNA content?

It is a minor observation that was included by way of providing a complete description of cell phenotype.

- p 17, "suggesting that ... loss-of-function", I do no not understand what this is based upon.

We show that the fitness of a *pbp1A* deletion is indistinguishable from the fitness of one of the *pbp1A* point mutants. This fact establishes that the point mutation had the same effects as a gene deletion thus supporting the claim that the point mutations identified during the course of the selection experiment decrease (or destroy) PBP1A function.

- p 25, at the top of the page: do you have a reference for the statement that a disorganized cell wall architecture is suited to the topology of spherical cells?

The statement is a conclusion that comes from our reasoning. It stems from the fact that it is impossible to entirely map the surface of a sphere with parallel strands.